# HIF-1α induces glycolytic reprograming in tissue-resident alveolar macrophages to promote cell survival during acute lung injury

**Parker S Woods[1], Lucas M Kimmig[1], Kaitlyn A Sun[1], Angelo Y Meliton[1], Obada R Shamaa[1], Yufeng Tian[1], Rengül Cetin-Atalay[1], Willard W Sharp[2], Robert B Hamanaka[1], Gökhan M Mutlu[1]\***

[1]Department of Medicine, Section of Pulmonary and Critical Care Medicine, The University of Chicago, Chicago, United States; [2]Department of Medicine, Section of Emergency Medicine, The University of Chicago, Chicago, United States

**\*For correspondence:**
gmutlu@medicine.bsd.uchicago.edu

**Abstract** Cellular metabolism is a critical regulator of macrophage effector function. Tissue-resident alveolar macrophages (TR-AMs) inhabit a unique niche marked by high oxygen and low glucose. We have recently shown that in contrast to bone marrow-derived macrophages (BMDMs), TR-AMs do not utilize glycolysis and instead predominantly rely on mitochondrial function for their effector response. It is not known how changes in local oxygen concentration that occur during conditions such as acute respiratory distress syndrome (ARDS) might affect TR-AM metabolism and function; however, ARDS is associated with progressive loss of TR-AMs, which correlates with the severity of disease and mortality. Here, we demonstrate that hypoxia robustly stabilizes HIF-1α in TR-AMs to promote a glycolytic phenotype. Hypoxia altered TR-AM metabolite signatures, cytokine production, and decreased their sensitivity to the inhibition of mitochondrial function. By contrast, hypoxia had minimal effects on BMDM metabolism. The effects of hypoxia on TR-AMs were mimicked by FG-4592, a HIF-1α stabilizer. Treatment with FG-4592 decreased TR-AM death and attenuated acute lung injury in mice. These findings reveal the importance of microenvironment in determining macrophage metabolic phenotype and highlight the therapeutic potential in targeting cellular metabolism to improve outcomes in diseases characterized by acute inflammation.

## Editor's evaluation

This work adds to an already abundant literature demonstrating that TR-AMs are a phenotypically and functionally distinct population of macrophages. Work in this article is the first to characterize the effect of hypoxia on metabolic and inflammatory responses in TR-AMs vs macrophages derived from other sites (bone marrow).

## Introduction

Glycolytic metabolism has been ascribed a central role in macrophage inflammatory processes (*Tannahill et al., 2013*; *Freemerman et al., 2014*; *Palsson-McDermott et al., 2015*; *Xie et al., 2016*; *Ip et al., 2017*). Much of our current understanding of this phenomenon has been elucidated in bone marrow-derived macrophages (BMDMs) and macrophage cell lines (i.e., THP-1 and RAW 264.7), which model macrophages of monocytic lineage. Considerably less is known about how other factors like local microenvironment and developmental origin may influence macrophage metabolic function.

Tissue-resident alveolar macrophages (TR-AMs) reside within the lumen of the alveolus where they are critical in maintaining lung homeostasis within the healthy airway and are the first responders to airborne pathogens and pollutants (*Hussell and Bell, 2014*). The alveolus maintains the highest oxygen concentration of any tissue compartment within the human body (*Carreau et al., 2011*). Moreover, under steady-state conditions, glucose concentrations within the airway lumen are less than one-tenth of blood glucose concentrations (*Baker et al., 2007*). These environmental conditions alone suggest a requirement for oxidative metabolism for cells that reside within the alveoli. Several studies have demonstrated that the unique characteristics of the alveolar microenvironment heavily influence macrophage function and immunometabolism (*Lavin et al., 2014*; *Svedberg et al., 2019*; *McQuattie-Pimentel et al., 2021*). Our group has recently demonstrated that TR-AMs rely predominantly on oxidative phosphorylation under steady-state conditions and that glycolysis is dispensable for the proinflammatory effector function in these cells (*Woods et al., 2020*). Together, these findings highlight the lung microenvironment's central role in dictating TR-AM responses.

Conditions associated with severe airway inflammation (i.e., acute respiratory distress syndrome [ARDS]) increase alveolar epithelial/endothelial barrier permeability (*Ware and Matthay, 2000*). This results in flooding of alveoli with fluid and recruitment of non-resident immune cells leading to severe local hypoxia as well as an increase in alveolar glucose levels (*Fröhlich et al., 2013*; *Campbell et al., 2014*; *Baker and Baines, 2018*). These abrupt changes in the alveolar microenvironment under a diseased state such as ARDS likely necessitate metabolic adaptation by TR-AMs in order to ensure optimal cellular fitness. Acute lung injury/ARDS is associated with a decline in the number of TR-AMs and the degree of TR-AM loss correlates with clinical outcomes (i.e., mortality) (*Fan and Fan, 2018*). Moreover, experimental depletion of TR-AMs results in an increase in the severity of acute lung injury and mortality (*Beck-Schimmer et al., 2005*; *Kim et al., 2008*; *Jaworska et al., 2014*; *Machado-Aranda et al., 2014*; *Nelson et al., 2014*; *Schneider et al., 2014*; *Cardani et al., 2017*). Whether or not changes in the alveolar microenvironment play a role in TR-AM cell death during acute lung injury/ARDS has yet to be explored.

Hypoxia-inducible factor 1-alpha (HIF-1α) is the most extensively characterized transcription factor responsible for cellular adaptation to low oxygen levels (*Semenza, 2012*). Under normoxia, oxygen-dependent proline hydroxylases prevent HIF-1α activation by marking it for proteasomal degradation. Conversely, under hypoxia, decreased hydroxylase activity promotes HIF-1α protein stabilization and translocation to the nucleus allowing for transcriptional responses to low oxygen levels, such as enhanced expression of genes related to glycolysis and angiogenesis. HIF-1α has been well characterized in macrophages of monocytic origin and identified to play a key role in macrophage infiltration and proinflammatory responses (*Cramer et al., 2003*; *Peyssonnaux et al., 2005*; *Tannahill et al., 2013*; *Matak et al., 2015*; *Palsson-McDermott et al., 2015*). However, little is known about the role that HIF-1α plays in TR-AM effector function and metabolism. In a study focusing on TR-AM development, the expression of HIF-1α and its target genes was found to be turned off following birth and this process was required for TR-AM maturation and normal effector function (*Izquierdo et al., 2018*). It is unknown, however, whether HIF-1α plays a role in TR-AM effector function after maturation or how local hypoxia and changes in the expression of HIF-1α might regulate the adaptation of mature TR-AMs to hypoxia or affect their effector function during acute lung injury.

To answer these questions, we used a variety of metabolic and immunological approaches. We observed that HIF-1α was undetectable in primary TR-AMs cultured under normoxia, but was robustly stabilized under hypoxia in a dose-dependent fashion. Upon hypoxic HIF-1α stabilization, TR-AMs acquired a glycolytic phenotype, which was not observed under normoxic conditions. In contrast, BMDMs exhibited no alterations in HIF-1α stabilization or glycolytic output in response to hypoxia. Analysis of lipopolysaccharide (LPS)-induced TR-AM metabolite signatures revealed significant increases in glycolytic intermediates under hypoxia. Hypoxia also altered the TR-AM cytokine profile in response to LPS and was able rescue the ETC inhibitor-induced impairment in cytokine production in TR-AMs.

Using influenza infection in mice to model acute lung injury, we found that TR-AM cell number decreased over the course of acute lung injury. Surviving TR-AMs exhibited a glycolytic gene signature supporting hypoxic adaptation. To determine how the ability to adapt to hypoxia affects TR-AM survival and function, we treated influenza-infected mice intratracheally with FG-4592, a HIF-1α stabilizer, which mimics hypoxic adaptation. Compared to control mice, FG-4592 prevented the loss of

TR-AMs, reduced lung injury, and increased survival. Collectively, our data suggest that HIF-1α plays a critical role in TR-AM metabolic adaptation to altered environmental conditions during acute lung injury. Promoting hypoxic adaptation and glycolytic metabolism in TR-AMs enables them to adapt to and survive the changes in the microenvironment during lung injury and consequently reduces lung inflammation and may offer a viable therapeutic strategy in treating ARDS arising from influenza or other severe viral infections, including COVID-19.

## Results

### TR-AMs exhibit HIF-1α stabilization and develop a glycolytic phenotype in response to hypoxia

We have recently shown that TR-AMs maintain a very low glycolytic rate that is not augmented by activation of inflammatory responses (*Woods et al., 2020*). Since TR-AMs inhabit an environment with high oxygen levels, we hypothesized that TR-AMs may not be able to induce glycolytic reprogramming in response to either inflammatory stimuli or to physiological hypoxia. Glycolysis stress tests were performed following overnight (16 hr) exposure to decreasing levels of ambient oxygen. Unlike their inability to induce glycolysis after inflammatory stimulus under normoxia, TR-AMs exhibited a progressive increase in the extracellular acidification rate (ECAR) in response to escalating degrees of ambient hypoxia (*Figure 1A*). Both basal rate of glycolysis and glycolytic reserve increased substantially when oxygen levels were lowered to 3.0 and 1.5% (*Figure 1B*). HIF-1α levels were nearly undetectable under normoxic conditions; however, with increasing degrees of hypoxia, HIF-1α stabilization occurred in a dose-dependent fashion and was detectable in the nucleus (*Figure 1C*; *Figure 1—source data 1*). Pretreating TR-AMs prior to hypoxia with echinomycin, an inhibitor of HIF-1α DNA binding activity (*Kong et al., 2005*), disrupted hypoxia-induced increases in glycolytic rate in a dose-dependent fashion (*Figure 1D*). Echinomycin also reduced hypoxia-induced increases in glycolytic enzyme protein expression (HK2 and LDH), suggesting that HIF-1α is required for glycolytic adaption to hypoxia in TR-AMs (*Figure 1E*; *Figure 1—source data 2*). In concurrence with the echinomycin data, siRNA knockdown of HIF-1α attenuated the hypoxia-induced increase in protein expression of glycolytic enzymes and lactate production in TR-AMs (*Figure 1—figure supplement 1A–C*, *Figure 1—figure supplement 1—source data 1*, *Figure 1—figure supplement 1—source data 2*).

Both short-term (2 hr) and prolonged (16 hr) exposure to hypoxia (1.5% $O_2$) led to significant increases in nuclear HIF-1α protein levels in TR-AMs (*Figure 1—figure supplement 2A*, *Figure 1—figure supplement 2—source data 1*). Glycolysis stress tests demonstrated that short-term hypoxia treatment failed to induce significant alterations in glycolysis or glycolytic capacity in TR-AMs compared to prolonged hypoxia treatment, suggesting that transcription and translation of glycolytic genes that are targets of HIF-1α are required (*Figure 1—figure supplement 2B and C*). Taken together, these data indicate that TR-AM HIF-1α stabilization in response to hypoxia is dose-dependent, and that prolonged hypoxia, but not short-term hypoxia exposure, leads to a functional glycolytic phenotype in TR-AMs.

### BMDMs have limited metabolic adaptation to hypoxia

Several studies have examined the effects of hypoxia on BMDM metabolism; however, they focused heavily on transcriptional changes in glycolytic gene expression as opposed to functional changes in glycolysis (*Bosco et al., 2006*; *Roiniotis et al., 2009*; *Delprat et al., 2020*). We found that, unlike TR-AMs, BMDMs exposed to hypoxia (16 hr) exhibit minimal changes in glycolytic rate or glycolytic capacity (*Figure 1F and G*). Interestingly, we found that BMDMs have high basal levels of nuclear HIF-1α protein under normoxic conditions and that HIF-1α expression in BMDMs did not significantly change in response to hypoxia as low as 1.5% $O_2$ (*Figure 1H*, *Figure 1—source data 3*). Echinomycin had minimal effect on the glycolytic output of hypoxic BMDMs (*Figure 1I*). Likewise, neither hypoxia nor hypoxia in combination with echinomycin altered glycolytic enzyme protein expression (HK2 and LDHA) in BMDMs (*Figure 1J*, *Figure 1—source data 4*). SiRNA knockdown of *Hif1a* also had no impact on glycolytic protein expression or lactate production in BMDMs (*Figure 1—figure supplement 1D–F*, *Figure 1—figure supplement 1—source data 3*, *Figure 1—figure supplement 1—source data 4*). Duration of hypoxia exposure (2 hr vs. 16 hr) had no significant effect on HIF-1α stabilization (*Figure 1—figure supplement 2D*, *Figure 1—figure supplement 2—source data 2*) or

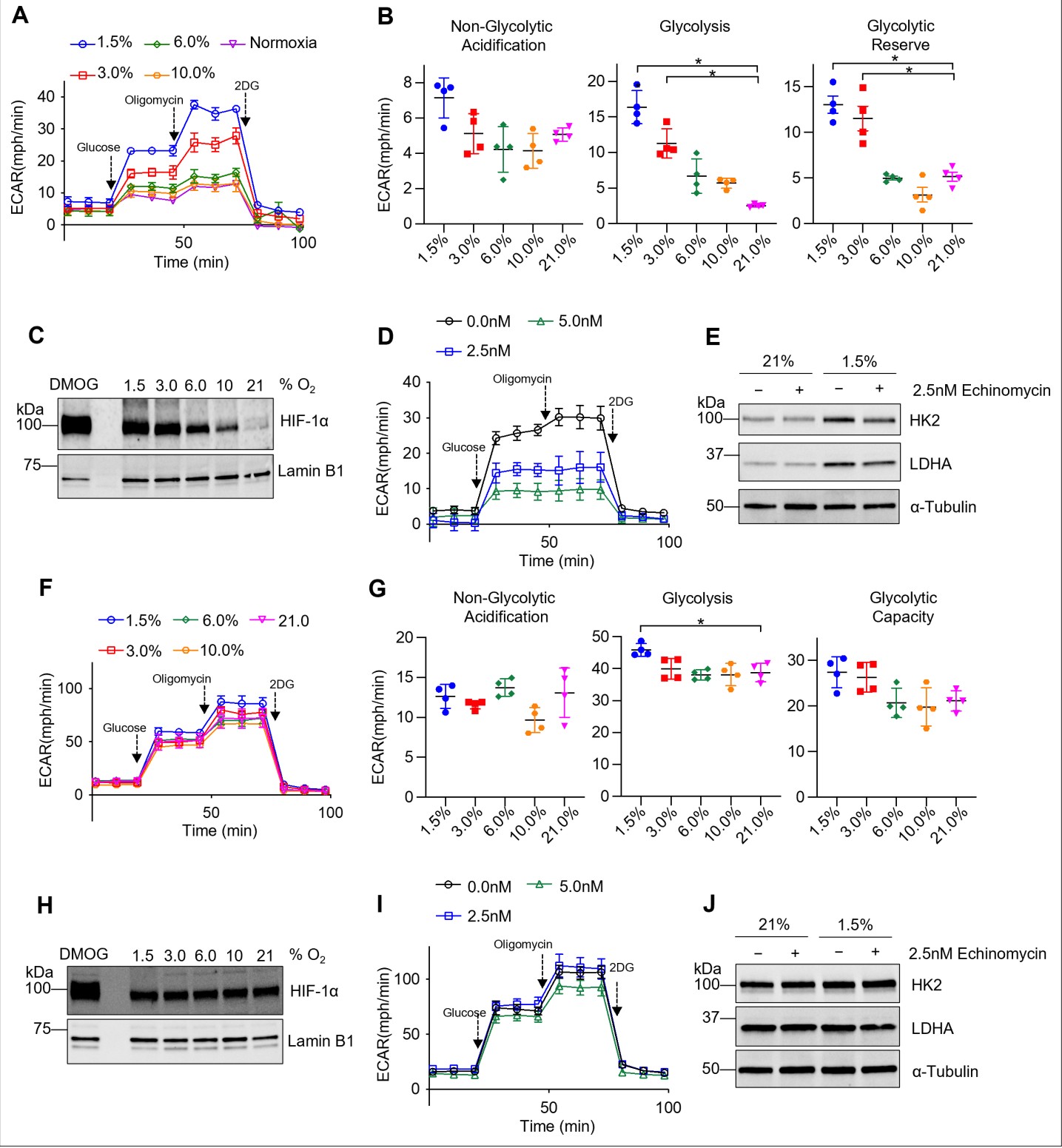

**Figure 1.** Tissue-resident alveolar macrophages (TR-AMs) exhibit hypoxia-inducible factor 1-alpha (HIF-1α) stabilization and develop a glycolytic phenotype in response to hypoxia, while bone marrow-derived macrophages (BMDMs) have limited metabolic adaptation to hypoxia. TR-AMs (**A–E**) and BMDMs (**F–J**) were incubated overnight (16 hr) at varying O₂ concentrations. (**A**) Using Seahorse XF24 analyzer, glycolysis was measured as extracellular acidification rate (ECAR). TR-AMs were sequentially treated with glucose, oligomycin (ATP synthase inhibitor), and 2-deoxyglucose (2-DG) (inhibitor of hexokinase 2, or glycolysis). (**B**) Interleaved scatter plots quantifying glycolytic parameters. Data represent at least three independent experiments (n = 4 separate wells per group). Glycolytic parameters were compared against 21% O₂ and significance was determined by one-way ANOVA with Bonferroni

*Figure 1 continued on next page*

*Figure 1 continued*

correction. (**C**) Western blot analysis of nuclear extract to assess HIF-1α expression in TR-AMs treated with different concentrations of $O_2$. DMOG served as a positive control. (**D**) Glycolysis stress test of TR-AMs under 1.5% $O_2$ in combination with echinomycin (16 hr). (**E**) Western blot analysis of whole-cell lysates of TR-AMs treated with 21 or 1.5% $O_2$ in combination with echinomycin (16 hr). (**F**) BMDM glycolysis measurements (ECAR) using Seahorse XF24 analyzer. (**G**) Interleaved scatter plots quantifying glycolytic parameters. Data represent at least three independent experiments (n = 4 separate wells per group). Glycolytic parameters were compared against 21% $O_2$ and significance was determined by one-way ANOVA with Bonferroni correction. (**H**) Western blot analysis of nuclear extract to assess HIF-1α expression in BMDMs treated with different concentrations of $O_2$. (**I**) Glycolysis stress test of BMDMs under 1.5% $O_2$ in combination with echinomycin (16 hr). (**J**) Western blot analysis of whole-cell lysates of BMDMs treated with 21 or 1.5% $O_2$ in combination with echinomycin (16 hr). All error bars denote mean ± SD. *$p<0.05$.

The online version of this article includes the following source data and figure supplement(s) for figure 1:

**Source data 1.** The effect of different $O_2$ concentrations on hypoxia-inducible factor 1-alpha HIF-1α expression in tissue-resident alveolar macrophages (TR-AMs).

**Source data 2.** The effect of echinomycin on glycolytic enzyme protein expression in tissue-resident alveolar macrophages (TR-AMs).

**Source data 3.** The effect of different $O_2$ concentrations on hypoxia-inducible factor 1-alpha (HIF-1α) expression in bone marrow-derived macrophages (BMDMs).

**Source data 4.** The effect of echinomycin on glycolytic enzyme protein expression in bone marrow-derived macrophages (BMDMs).

**Figure supplement 1.** Knockdown of *Hif1a* diminishes hypoxia-induced glycolytic phenotype in tissue-resident alveolar macrophages (TR-AMs).

**Figure supplement 1—source data 1.** Validation of *Hif1a* siRNA knockdown in tissue-resident alveolar macrophages (TR-AMs).

**Figure supplement 1—source data 2.** The effect of *Hif1a* siRNA knockdown on glycolytic enzyme protein expression in tissue-resident alveolar macrophages (TR-AMs) under normoxia and hypoxia.

**Figure supplement 1—source data 3.** Validation of *Hif1a* siRNA knockdown in bone marrow-derived macrophages (BMDMs).

**Figure supplement 1—source data 4.** The effect of *Hif1a* siRNA knockdown on glycolytic enzyme protein expression in bone marrow-derived macrophages (BMDMs) under normoxia and hypoxia.

**Figure supplement 2.** Prolonged but not short-term hypoxia induces glycolysis in tissue-resident alveolar macrophages (TR-AMs).

**Figure supplement 2—source data 1.** Expression of hypoxia-inducible factor 1-alpha (HIF-1α) protein in tissue-resident alveolar macrophages (TR-AMs) at different time points following exposure to hypoxia.

**Figure supplement 2—source data 2.** Expression of hypoxia-inducible factor 1-alpha (HIF-1α) protein in bone marrow-derived macrophages (BMDMs) at different time points following exposure to hypoxia.

glycolysis (*Figure 1—figure supplement 2E and F*) in BMDMs. Collectively, these data demonstrate that hypoxia has minimal effect on glycolytic function and HIF-1α stabilization in BMDMs.

## The hypoxia-induced transcriptomic response differs substantially between TR-AMs and BMDMs

To better understand the observed differences in hypoxia-induced glycolytic metabolism between TR-AMs and BMDMs, we performed RNA-sequencing to assess global alterations in gene expression. We found 741 DEGs (512 upregulated and 229 downregulated) in TR-AMs in response to 1.5% $O_2$ compared to only 260 DEGs (214 upregulated and 46 downregulated) in BMDMs (*Figure 2A*). Reactome pathway analysis revealed that hypoxia altered a large number of TR-AM genes in multiple pathways ranging from cellular metabolism, hemostasis, and immune cell function, while the majority of BMDM genes affected by hypoxia were related to carbohydrate metabolism (*Figure 2B*). Hypoxia led to the most significant increases in the expression of glycolytic and other HIF-1α target genes in both TR-AMs and BMDMs. These same genes were significantly lower in TR-AMs compared to BMDMs under normoxic conditions (*Figure 2C*, *Figure 2—source data 1*). This is in direct agreement with our previous findings (*Woods et al., 2020*). A side-by-side comparison demonstrated that the level of HIF-1α expression in hypoxic TR-AMs is similar to that of BMDMs under normoxia and hypoxia (*Figure 2D*, *Figure 2—source data 2*). Hypoxia-induced HIF-1α expression in TR-AMs correlated with increase in glycolytic (HK2 and LDHA) and prolyl hydroxylase (EGLN1 and EGLN3) protein expression in TR-AMs (*Figure 2E*, *Figure 2—source data 3*). This was not the case in BMDMs in which hypoxia exposure did not alter protein expression of glycolytic genes. These results demonstrate that hypoxia induces transcriptomic alterations in TR-AMs that lead to changes in protein expression and metabolic function. In contrast, BMDMs exhibit high levels of HIF-1α and HIF-1α target proteins at baseline and

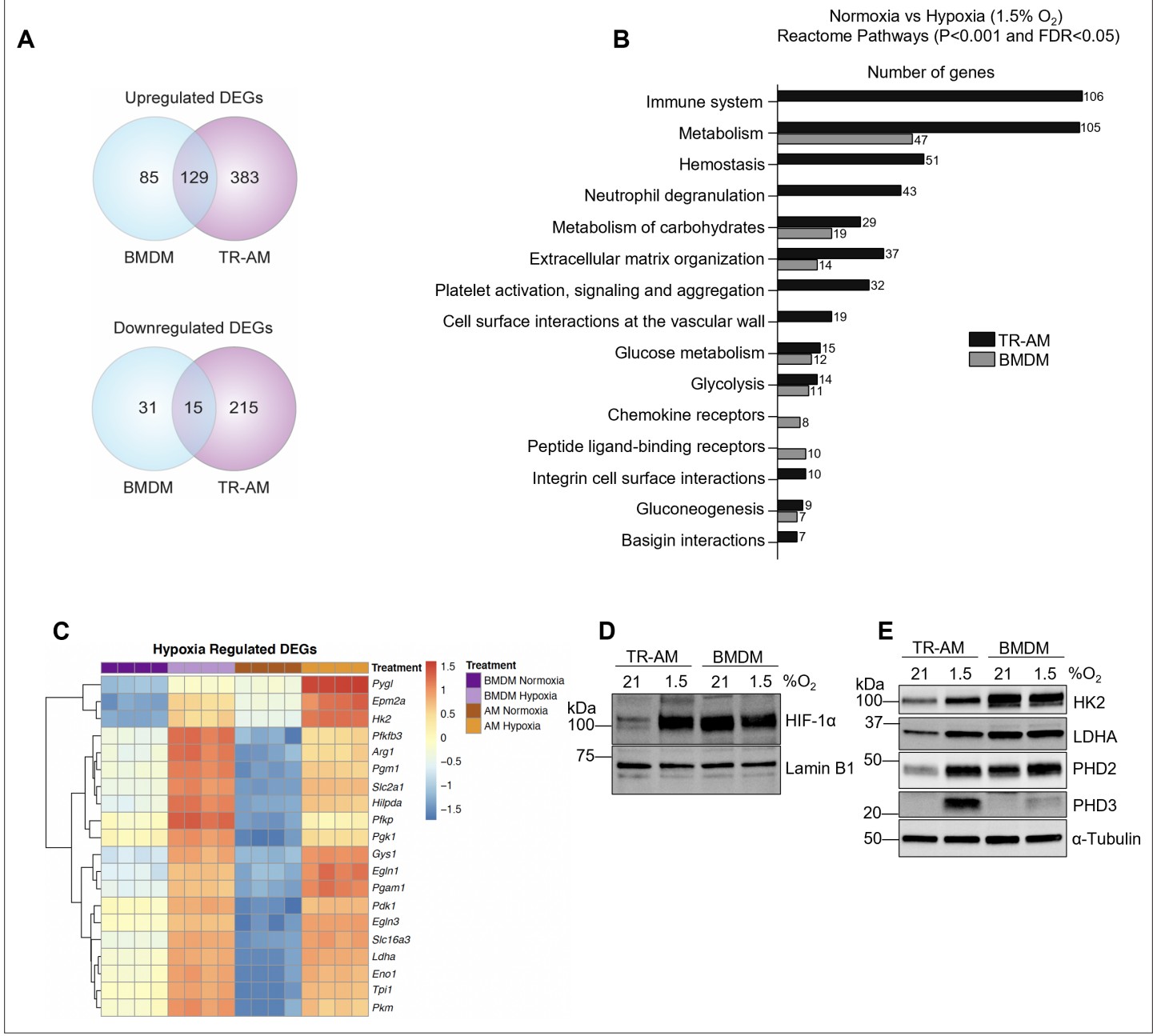

**Figure 2.** The hypoxia-induced transcriptomic response differs substantially between tissue-resident alveolar macrophages (TR-AMs) and bone marrow-derived macrophages (BMDMs). TR-AMs and BMDMs were incubated overnight (16 hr) under normoxia (21.0% $O_2$) or hypoxia (1.5% $O_2$). (**A**) Venn diagrams show differentially expressed genes (DEGs) altered by hypoxia in TR-AMs (741 total DEGs), and BMDMs (260 total DEGs). DEGs were identified using DESeq2 at FC > 2 and false discovery rate (FDR)-adjusted p-value of <0.05. (**B**) Reactome pathway enrichment comparing number of genes in a given pathway altered by hypoxia in TR-AMs and BMDMs. (**C**) Heatmap representing the top 20 significant metabolic genes altered by hypoxia in both TR-AMs and BMDMs. (**D**) Western blot analysis of nuclear extracts to assess hypoxia-inducible factor 1-alpha (HIF-1α) protein expression. (**E**) Western blot analysis of whole cell extracts to assess glycolytic enzyme (HK2, LDH) and prolyl hydroxylase (PHD2, PHD3) protein expression.

The online version of this article includes the following source data for figure 2:

**Source data 1.** Read count data for hypoxia-regulated genes in tissue-resident alveolar macrophages (TR-AMs) and bone marrow-derived macrophages (BMDMs).

**Source data 2.** Differences in hypoxia-inducible factor 1-alpha (HIF-1α) expression between tissue-resident alveolar macrophages (TR-AMs) and bone marrow-derived macrophages (BMDMs) under normoxia and hypoxia.

*Figure 2 continued on next page*

*Figure 2 continued*

**Source data 3.** Differences in glycolytic enzyme and prolyl hydroxylase protein expression between tissue-resident alveolar macrophages (TR-AMs) and bone marrow-derived macrophages (BMDMs) under normoxia and hypoxia.

do not further increase the expression of these proteins in response to hypoxia, despite a hypoxia-adaptive mRNA transcription (*Figure 2D and E*).

## Hypoxia modulates TR-AM cytokine production and metabolic response to LPS

Hypoxia and HIF-1α are thought to be central to the inflammatory response of macrophages (*Cramer et al., 2003*; *Tannahill et al., 2013*; *Palsson-McDermott et al., 2015*). To determine the effect of HIF-1α stabilization on TR-AM's effector response, we measured the production of proinflammatory cytokines in response to LPS under hypoxia. TR-AMs were exposed overnight to hypoxia (1.5% $O_2$) or normoxia and then subsequently treated with LPS while maintaining original $O_2$ conditions. Hypoxia alone did not stimulate cytokine production without LPS treatment. Hypoxic TR-AMs secreted significantly higher levels of TNF-α, KC, and IL-1β in response to LPS compared to normoxic controls. In contrast, IL-6 and CCL2 secretion was decreased in hypoxic TR-AMs (*Figure 3A*). The cytokine gene expression pattern in hypoxic TR-AMs treated with LPS mirrored the secreted cytokine profile (*Figure 3B*). Moreover, enhanced proIL-1β protein production was observed in hypoxic TR-AMs treated with LPS (*Figure 3C*, *Figure 3—source data 1*). While BMDMs experienced limited metabolic alterations in response to hypoxia, treatment with LPS revealed that hypoxia induced similar alterations in their cytokine profile. Hypoxic BMDMs had increased TNF-α, KC, and IL-1β, and decreased IL-6 secretion (*Figure 3—figure supplement 1*). The only discordance in the hypoxic cytokine response profile between TR-AMs and BMDMs was CCL2, which remained unchanged in hypoxic BMDMs in response to LPS compared to normoxic controls (*Figure 3—figure supplement 1*).

We and others have shown that BMDMs exhibit an immediate enhancement in glycolytic output in response to LPS (*Figure 3—figure supplement 2*). It is thought that this increase in glycolysis following LPS supports the proinflammatory response. We have shown that LPS-induced inflammation in TR-AMs is independent of glycolysis, including the rise in glycolysis following LPS injection (*Woods et al., 2020*). Given that hypoxia elevated HIF-1α levels and glycolytic rates in TR-AMs, we sought to determine whether hypoxia could alter TR-AM glycolytic responsiveness to LPS. We found that despite the fact that hypoxia increased the glycolytic rate of TR-AMs at baseline, TR-AM glycolysis remained unresponsive to LPS injection (*Figure 3D*). Using capillary electrophoresis-mass spectrometry to measure glycolytic metabolite levels, we found that consistent with increased glycolytic output after hypoxia, levels of glycolytic intermediate metabolites (glucose-6 phosphate, fructose 1,6 diphosphate, glycerol 3-phosphate, dihydroxyacetone phosphate) and lactate were increased in response to hypoxia alone. Interestingly, hypoxic TR-AMs exhibited further increases in glycolytic intermediates in the presence of LPS (6 hr) compared to normoxic cells, suggesting that while LPS increases cellular levels of glycolytic metabolites in hypoxic TR-AMs, this does not manifest as acute lactate secretion (*Figure 3E*). These data demonstrate that hypoxia leads to significant alterations in TR-AM cytokine production and increased glycolytic metabolites in response to prolonged LPS treatment. However, unlike the prototypical BMDM response, hypoxic TR-AMs do not immediately increase their extracellular acidification in response to LPS.

We have previously shown that unlike BMDMs, TR-AM's effector function is acutely sensitive to mitochondrial inhibition (*Woods et al., 2020*). Since TR-AM capacity for glycolysis expands with decreasing levels of $O_2$, we next sought to assess mitochondrial function under hypoxia and performed a mitochondrial stress test on TR-AMs that had been exposed to varying oxygen concentrations. Interestingly, mild-to-moderate degrees of ambient hypoxia did not appear to significantly alter overall mitochondrial function in these cells. Only 1.5% $O_2$ caused significant reductions in oxygen consumption rate (OCR) across all mitochondrial parameters (*Figure 4A and B*). ECAR tracings during the mitochondrial stress test demonstrated that, other than severe hypoxia (1.5% $O_2$), the majority of acid produced under mildto moderate hypoxia is derived from $CO_2$, as the application of rotenone and antimycin A led to a significant reduction in ECAR (*Figure 4C*). When exposed to 1.5% $O_2$, TR-AMs' energy is derived mostly from glycolysis with no significant contribution of mitochondrial-derived $CO_2$ to extracellular acidification. Overall BMDM mitochondrial function was impaired by 1.5% $O_2$, but the

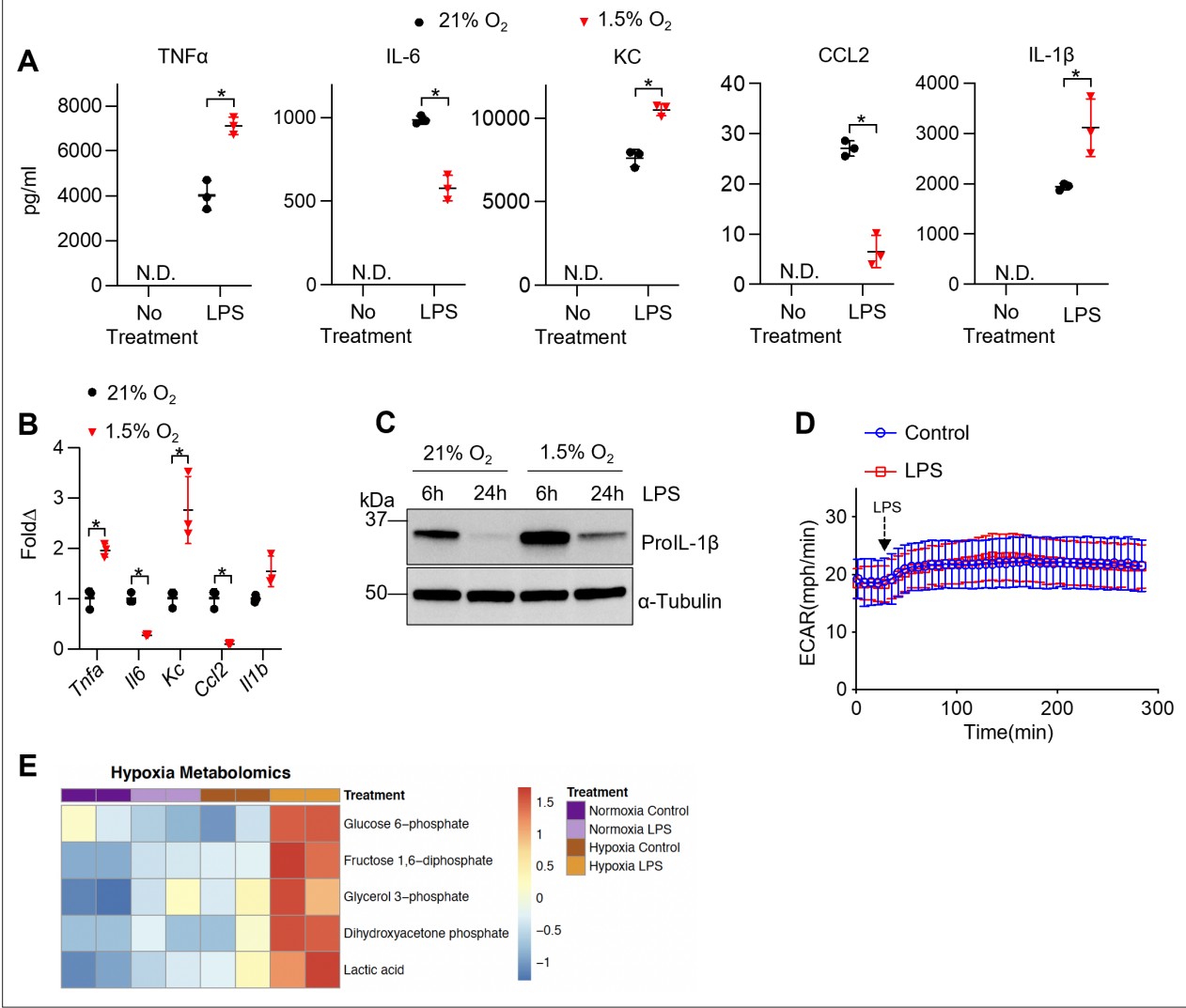

**Figure 3.** Hypoxia modulates tissue-resident alveolar macrophage (TR-AM) cytokine production and metabolic response to lipopolysaccharide (LPS). TR-AMs were incubated overnight (16 hr) under 21 or 1.5% O₂, then stimulated with 20 ng/ml LPS for 6 hr while maintaining pretreatment conditions. For IL-1β measurements, 5 mM ATP was added to TR-AMs for 30 min following 6 hr LPS treatment to activate caspase 1, ensuring IL-1β release. (**A**) We measured cytokine (TNFα, IL-6, KC, CCL2, and IL-1β) levels in media using ELISA. Data represent at least three independent experiments; n = 3 per group. Significance was determined by unpaired, two-tailed *t*-test. (**B**) qPCR was used to measure mRNA expression (*Tnfa, Il6, Kc, Ccl2,* and *Il1b*). Gene expression was normalized to corresponding gene ct values in 21% group and represented as fold change using the ΔΔct method. Data represent at least three independent experiments; n = 3 per group. Significance was determined by unpaired, two-tailed *t*-test. (**C**) Western blot analysis of whole-cell extracts at 6 and 24 hr post LPS treatment. (**D**) Extracellular acidification rate (ECAR) was measured in following acute LPS injection (final concentration: 20 ng/ml) in TR-AMs conditioned in 1.5% O₂. (**E**) Capillary electrophoresis-mass spectrometry (CE-MS) metabolite heatmap for glycolytic intermediates. All error bars denote mean ± SD. *p<0.05.

The online version of this article includes the following source data and figure supplement(s) for figure 3:

**Source data 1.** Changes in lipopolysaccharide (LPS)-induced expression of proIL-1β protein in tissue-resident alveolar macrophages (TR-AMs) under normoxia and hypoxia.

**Figure supplement 1.** Hypoxia alters cytokine production in bone marrow-derived macrophages (BMDMs).

**Figure supplement 2.** Lipopolysaccharide (LPS) induces an immediate increase in glycolysis in bone marrow-derived macrophages (BMDMs).

effect was greatly diminished compared to TR-AMs (*Figure 4—figure supplement 1A and B*). BMDM ECAR tracing during mitochondrial stress test demonstrated that most acid production remained unchanged in response to rotenone and antimycin A regardless of O₂ concentration (*Figure 4—figure supplement 1C*). This suggests that BMDM acidification is glycolyticallyderived under both normoxia and hypoxia.

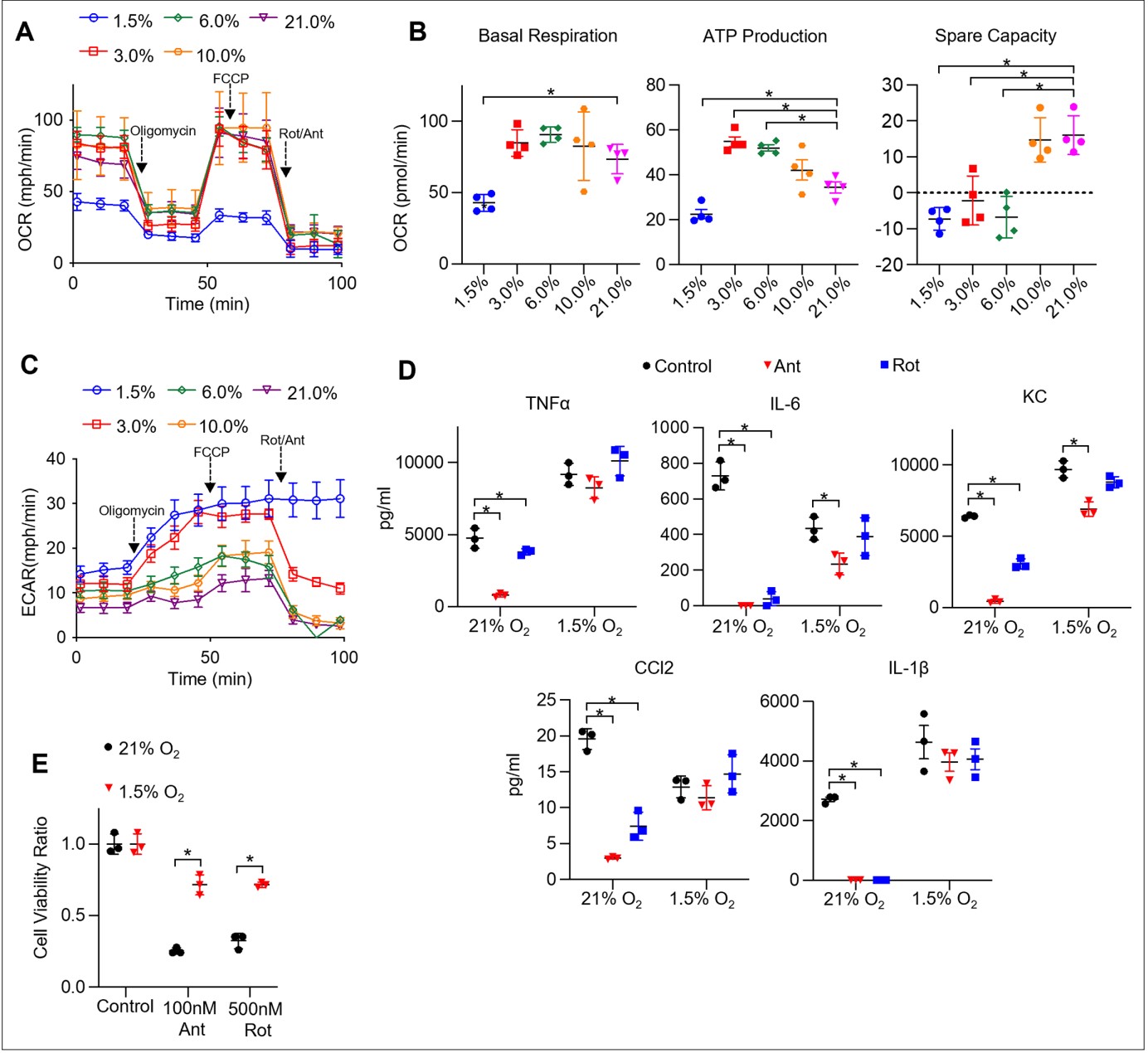

**Figure 4.** Hypoxia rescues ETC inhibitor-induced cell death and impaired cytokine production in tissue-resident alveolar macrophages (TR-AMs). (**A**) Mitochondrial stress test to measure oxygen consumption rate (OCR) using Seahorse XF24 in TR-AMs, which were treated sequentially with oligomycin (ATP synthase inhibitor), FCCP (uncoupler), and rotenone (Rot)/antimycin A (Ant) (complex I and III inhibitors, respectively). (**B**) Interleaved scatter plots quantifying mitochondrial respiration parameters. Data represents at least three experiments (n = 4 separate wells per group). Mitochondrial parameters were compared against 21% $O_2$ and significance was determined by one-way ANOVA with Bonferroni correction. (**C**) Extracellular acidification rate (ECAR) measurement during mitochondrial stress test to visualize TR-AMs' ability to upregulate glycolysis in response to mitochondrial inhibition. (**D**) TR-AMs were incubated overnight (16 hr) under 21 or 1.5% $O_2$, then stimulated with 20 ng/ml lipopolysaccharide (LPS) in the presence or absence of mitochondrial inhibitors (20 nM Ant or Rot) for 6 hr while maintaining pretreatment conditions. ELISA was used to measure secreted cytokine (TNFα, IL-6, KC, CCL2, and IL-1β) levels in media. ATP added to cells prior to collection for IL-1β assessment. Data represent at least three independent experiments; n = 3 per group. Significance was determined by one-way ANOVA with Bonferroni correction. (**E**) TR-AMs were cultured under 21 or 1.5% $O_2$ for 6 hr, then treated with mitochondrial inhibitors (100 nM Ant or 500 nM Rot) overnight and a sulforhodamine B assay was performed to measure cytotoxicity. Graphs represent cell viability compared to control, 21% $O_2$ group. Data represent at least three independent experiments (n = 3 per group). Significance was determined by two-way ANOVA with Bonferroni correction. All error bars denote mean ± SD. *p<0.05.

The online version of this article includes the following figure supplement(s) for figure 4:

**Figure supplement 1.** The effect of hypoxia on bone marrow-derived macrophage (BMDM) mitochondrial function, cytokine production, and cell viability under ETC inhibition.

TR-AM cytokine production in response to LPS was highly susceptible to inhibition by low doses of ETC inhibitors, rotenone and antimycin A, under normoxic conditions. This effect was greatly attenuated after exposure to hypoxia (*Figure 4D*). Additionally, high doses of ETC inhibitors induce cytotoxicity in normoxic TR-AMs, but hypoxic preconditioning significantly enhanced TR-AM cell viability (*Figure 4E*). In contrast, BMDM cytokine production was only marginally affected by ETC inhibition with the exception of observed decrease in IL-1β. Unlike TR-AMs, hypoxia did not significantly alter BMDM cytokine production in the presence of ETC inhibitors (*Figure 4—figure supplement 1D*). Similarly, ETC inhibition did not induce cytotoxicity in BMDMs under normoxia or hypoxia (*Figure 4— figure supplement 1E*).

## TR-AM survival correlates with a shift to glycolytic metabolism during influenza-induced acute lung injury

LPS is a well-known and potent macrophage activator that in isolation can be used to investigate essential immune functions, such as cytokine production, signal transduction, and immunometabolism. It induces a broad range of inflammatory effects in macrophages, making it a convenient tool to study overall immune fitness in vitro. However, in vivo studies have shown that LPS instillation into the murine airway leads to an immune response predominated by infiltrating neutrophils, making LPS-induced acute lung injury an unsuitable model to study macrophages (*Chignard and Balloy, 2000*). Compared to LPS, influenza infection is a more clinically relevant model of ARDS, and various macrophages populations play a larger role in both exacerbating and limiting lung injury in this model (*Short et al., 2014*). Several groups have demonstrated that TR-AMs undergo cell death in response to influenza infection and that depletion of TR-AMs is associated with worse outcomes in models of influenza-induced acute lung injury (*Kim et al., 2008*; *Jaworska et al., 2014*; *Nelson et al., 2014*; *Schneider et al., 2014*; *Cardani et al., 2017*). To confirm this phenomenon, we utilized PKH26 Red Fluorescent Linker dye to specifically label, track, and collect TR-AMs over the time course of infection as we and others have previously described (*Maus et al., 2001*; *Maus et al., 2002*; *Woods et al., 2020*). In agreement with *Zhu et al., 2021*, we found that there was a significant decrease in TR-AMs (PKH26+) at 3 (D3) and 6 (D6) days post infection (dpi) along with a subsequent increase in infiltrating, monocyte-derived alveolar macrophages (Mo-AMs) (*Figure 5A*). From these experiments, we performed RNAseq on sorted TR-AMs (PKH26+) and Mo-AMs (PKH26−) at D0, D3, and D6 (note: Mo-AMs are not present in an uninfected [D0] mouse) to identify changes in the metabolic gene signature of these two macrophage populations during influenza infection. From D0 to D6, RNAseq data revealed that TR-AMs experienced decreased expression in genes related to oxidative phosphorylation with simultaneous increased expression of genes related to glycolytic metabolism. Moreover, the metabolic gene signature of D6 TR-AMs was most similar to that of Mo-AMs at D3 and D6 (*Figure 5B and C*, *Figure 5—source data 1*, *Figure 5—source data 2*). Taken together, these data suggest that influenza-induced acute lung injury leads to a decrease in TR-AM number, and that the surviving TR-AMs' gene signature shifts away from genes related to mitochondrial metabolism in favor of glycolysis.

## HIF-1α stabilization increases TR-AM survival and improves outcomes in influenza-induced acute lung injury

Given that the reduced TR-AM population on D6 presented with a glycolytic gene signature, we hypothesized that TR-AM survival was dependent upon a metabolic shift to glycolysis. In other words, a decrease in TR-AM numbers overtime was due to a large fraction of the cells failing to metabolically adapt to the conditions of the infected/hypoxic alveoli. Thus, TR-AMs that could not adapt to hypoxia and retained primarily mitochondria-driven metabolism died off while TR-AMs that shifted to glycolytic metabolism survived. To test this hypothesis, we first sought to determine whether stabilization of HIF-1α was sufficient to induce a hypoxic metabolic state in AMs without altering $O_2$ levels. To do this, we treated cells with FG-4592, an inhibitor of HIF prolyl hydroxylases. FG-4592 has a greater potency and fewer off target effects compared to DMOG, which broadly inhibits 2-oxoglutarate-dependent oxygenases (*Singh et al., 2020*). TR-AMs treated with FG-4592 for 16 hr exhibited a significant dose-dependent increase in glycolysis (*Figure 6A and B*). FG-4592 induced robust HIF-1α stabilization leading to increased expression of HK2, LDHA, PHD1, and PHD3 (*Figure 6C and D*, *Figure 6— source data 1*, *Figure 6—source data 2*). Unlike hypoxia (1.5% $O_2$), FG-4592 had very little impact on

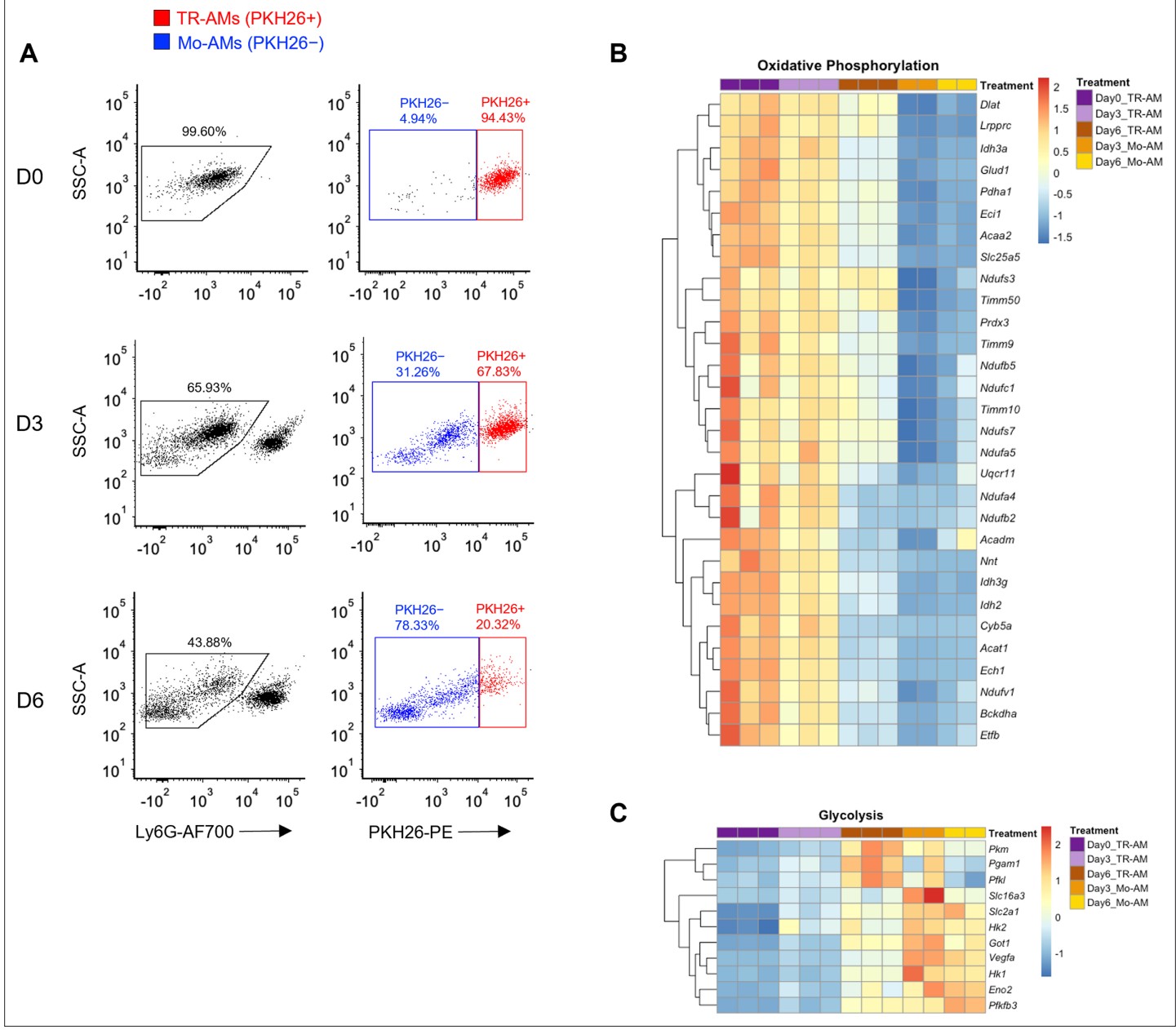

**Figure 5.** Tissue-resident alveolar macrophage (TR-AM) survival correlates with a shift to glycolytic metabolism during influenza-induced acute lung injury. (**A**) FACS plots of bronchoalveolar lavage fluid (BALF) samples collected from C57BL/6 mice infected with PR8 (100 PFU) at baseline (D0), 3 days (D3), and 6 days (D6) post infection. First, debris, red blood cells, and lymphocytes were eliminated based on size (forward scatter signal [FSC]) and granularity (side scatter signal [SSC]). Samples were first gated on single cells based on the SSC/FSC, and then live cells were selected (SYTOX Green−). Ly6G− used to exclude neutrophils. TR-AMs were identified as being PKH26+, and nonresident/infiltrating monocyte-derived alveolar macrophages (Mo-AMs) were PKH26−. Gene expression heatmaps representing (**B**) oxidative phosphorylation and (**C**) glycolytic gene expression. Heatmaps were generated through differentially expressed gene (DEG) analysis of UniProt oxidative phosphorylation and glycolysis gene sets for FAC TR-AMs (PKH+; n = 3/group) and Mo-AMs (PKH26−; n = 2/group) over the infection time course.

The online version of this article includes the following source data for figure 5:

**Source data 1.** Read counts data for genes of oxidative phosphorylation in tissue-resident alveolar macrophages (TR-AMs) and monocyte-derived alveolar macrophages (Mo-AMs).

**Source data 2.** Read counts data for genes of glycolysis in tissue-resident alveolar macrophages (TR-AMs) and monocyte-derived alveolar macrophages (Mo-AMs).

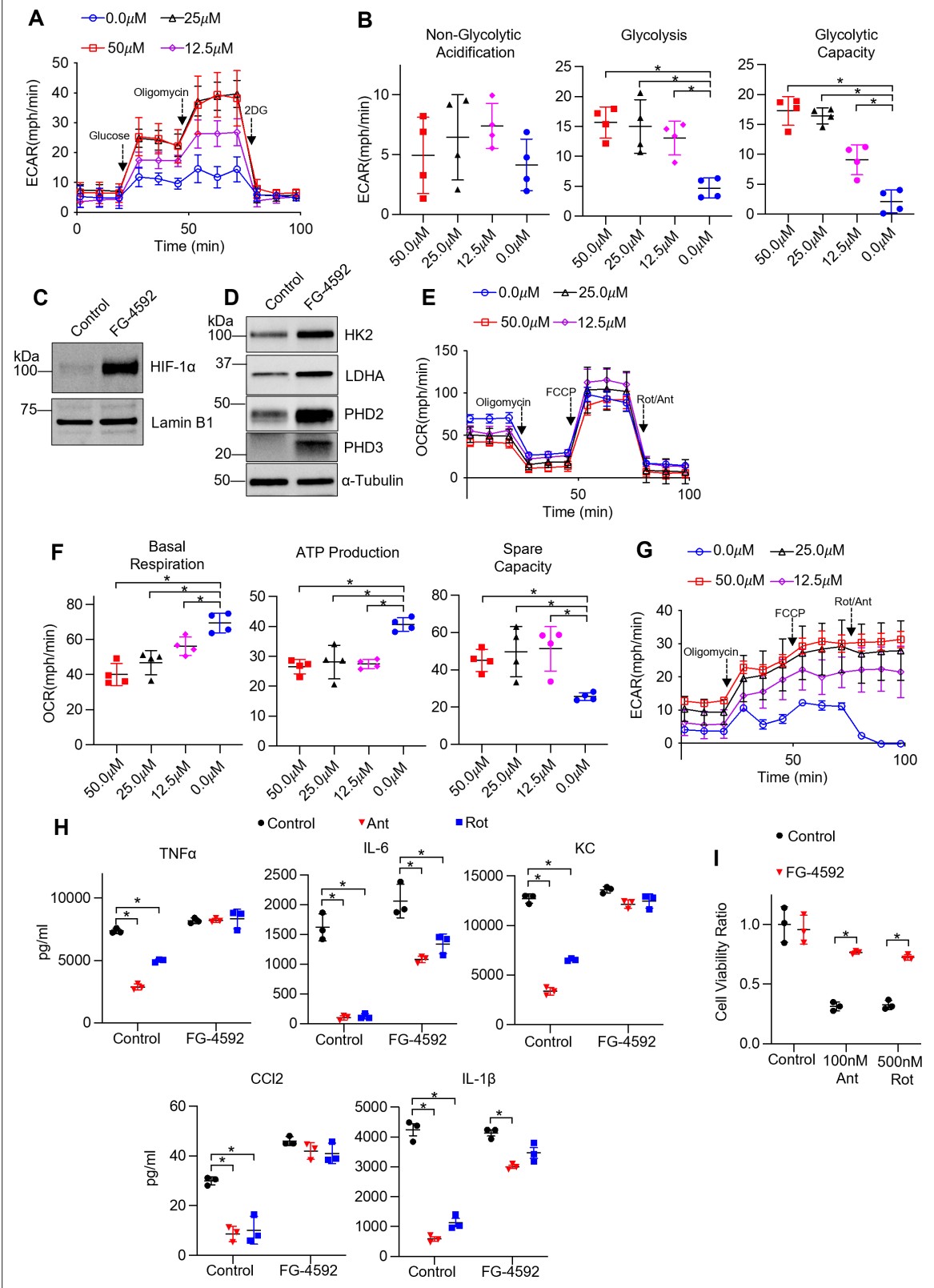

**Figure 6.** Non-hypoxic stabilization of hypoxia-inducible factor 1-alpha (HIF-1α) induces glycolysis and rescues ETC inhibitor-induced reduction in cytokine production and cell death in tissue-resident alveolar macrophages (TR-AMs). TR-AMs were treated (16 hr) overnight ±FG-4592 (25.0 μM when not stated otherwise). (**A**) Glycolysis was measured as extracellular acidification rate (ECAR). (**B**) Quantification of glycolytic parameters. Data represent at least three independent experiments (n = 4 separate wells per group). Glycolytic parameters compared to control group (0.0 μM) and

*Figure 6 continued on next page*

*Figure 6 continued*

significance was determined by one-way ANOVA with Bonferroni correction. (**C**) Western blot analysis of nuclear extract for HIF1α expression and (**D**) whole cell lysate for glycolytic enzyme and prolyl hydroxylase expression. (**E**) Mitochondrial stress test to measure oxygen consumption rate (OCR). (**F**) Quantification of mitochondrial respiration parameters. Data represents at least three experiments (n = 4 separate wells per group). Mitochondrial parameters were compared to control group (0.0 µM) and significance was determined by one-way ANOVA with Bonferroni correction. (**G**) ECAR measurement during mitochondrial stress test. (**H**) TR-AMs were pretreated overnight (16 hr) with 0.0 µM (no treatment) or 25.0 µM FG-4592, then stimulated with 20 ng/ml lipopolysaccharide (LPS) in the presence or absence of mitochondrial inhibitors (20 nM antimycin A [Ant] or rotenone [Rot]) for 6 hr while maintaining pretreatment conditions. Sandwich ELISA was used to measure secreted cytokine (TNFα, IL-6, KC, and CCL-2). Data represents at least three independent experiments; n = 3 per group. (**I**) TR-AMs were treated with FG-4592 for 6 hr, then treated with mitochondrial inhibitors (100 nM Ant or 500 nM Rot) overnight and a sulforhodamine B assay was performed to measure cytotoxicity. Bar graphs represent cytotoxicity compared to control, 0.0 µM group. Data represents at least three independent experiments (n = 3 per group). Significance was determined by two-way ANOVA with Bonferroni correction. All error bars denote mean ± SD. *p<0.05.

The online version of this article includes the following source data for figure 6:

**Source data 1.** The effect of FG-4592 on hypoxia-inducible factor 1-alpha (HIF-1α) expression in tissue-resident alveolar macrophages (TR-AMs).

**Source data 2.** The effect of FG-4592 on glycolytic enzyme and prolyl hydroxylase protein expression in tissue-resident alveolar macrophages (TR-AMs).

overall mitochondrial fitness in TR-AMs (*Figure 6E*). Basal respiration and mitochondrial ATP production were reduced, but spare mitochondrial compacity was increased, signaling a shift toward glycolytic ATP production at baseline but no loss in overall mitochondrial function (*Figure 6E–G*). Like hypoxia, FG-4592 treatment could also rescue ETC inhibitor-induced impairment in cytokine production (*Figure 6H*) and cell death in TR-AMs (*Figure 6I*). However, unlike TR-AMs exposed to 1.5% O$_2$, FG-4592 did not broadly alter LPS cytokine responses, suggesting that changes in the cytokine profile under hypoxia are oxygen-dependent, but remain independent of HIF-1α stabilization (*Figure 6H*).

We next treated mice intratracheally with one dose of FG-4592 at the time of infection to evaluate the effect of early glycolytic adaptation on TR-AM survival and influenza-induced acute lung injury. Strikingly, FG-4592 treatment resulted in increased TR-AM (PKH26+ cells) survival at 6 dpi compared to infected controls (*Figure 7A*). The increase in TR-AM survival in FG-4592-treated mice was associated with reduced alveolar permeability (*Figure 7B*). FG-4592 treatment also led to a reduction in pro-inflammatory cytokine levels within the alveolar space at 6 dpi (*Figure 7C*). Most importantly, FG-4592-treated mice experienced reduced weight loss and improved survival compared to infected controls (*Figure 7D and E*). Taken together, these data suggest that intratracheal FG-4592 treatment can increase TR-AM survival and improve outcomes in influenza-infected mice.

## Discussion

ARDS is associated with high morbidity and mortality. Despite many decades of research, treatment remains supportive and there is no therapy that directly targets the pathogenesis of ARDS. Infection is the main cause of ARDS. Respiratory viruses such as influenza A virus and SARS-CoV-2 cause significant mortality by causing ARDS. In fact, the 2009 influenza pandemic and the ongoing COVID-19 pandemic have shown that acute respiratory illnesses can have a profound effect on society in the 21st century. ARDS caused by influenza A and SARS-CoV-2 viruses is associated with loss of TR-AMs, which correlates with disease severity and mortality (*Ghoneim et al., 2013*; *Liao et al., 2020*; *Grant et al., 2021*; *Zhu et al., 2021*). Because TR-AMs maintain a central role in lung homeostasis and response to airborne pathogens, understanding basic TR-AM processes, like metabolic adaptation in response to an altered lung environment during acute lung injury/ARDS, may allow us to therapeutically rescue TR-AM cell death and augment their function to improve outcomes in acute respiratory illnesses.

We have previously demonstrated that TR-AMs are remarkably adapted to the high-oxygen, low-glucose environment of the alveolar lumen and do not require glucose or glycolysis to carry out their effector function as monocyte-derived macrophages do (*Woods et al., 2020*). ARDS leads to significant hypoxia to which TR-AMs need to adapt; thus, we sought to determine whether TR-AMs displayed metabolic plasticity during conditions of low oxygen. In this study, we found that TR-AMs adapted to hypoxia through HIF-1α induction. Hypoxia stabilized TR-AM HIF-1α in a dose-dependent manner, resulting in robust increases in glycolytic protein expression and function. These changes were dependent on HIF-1α as hypoxic TR-AMs treated with echinomycin, an inhibitor of HIF-1α DNA binding activity, lost their glycolytic capabilities. In contrast, BMDMs exhibited robust HIF-1α

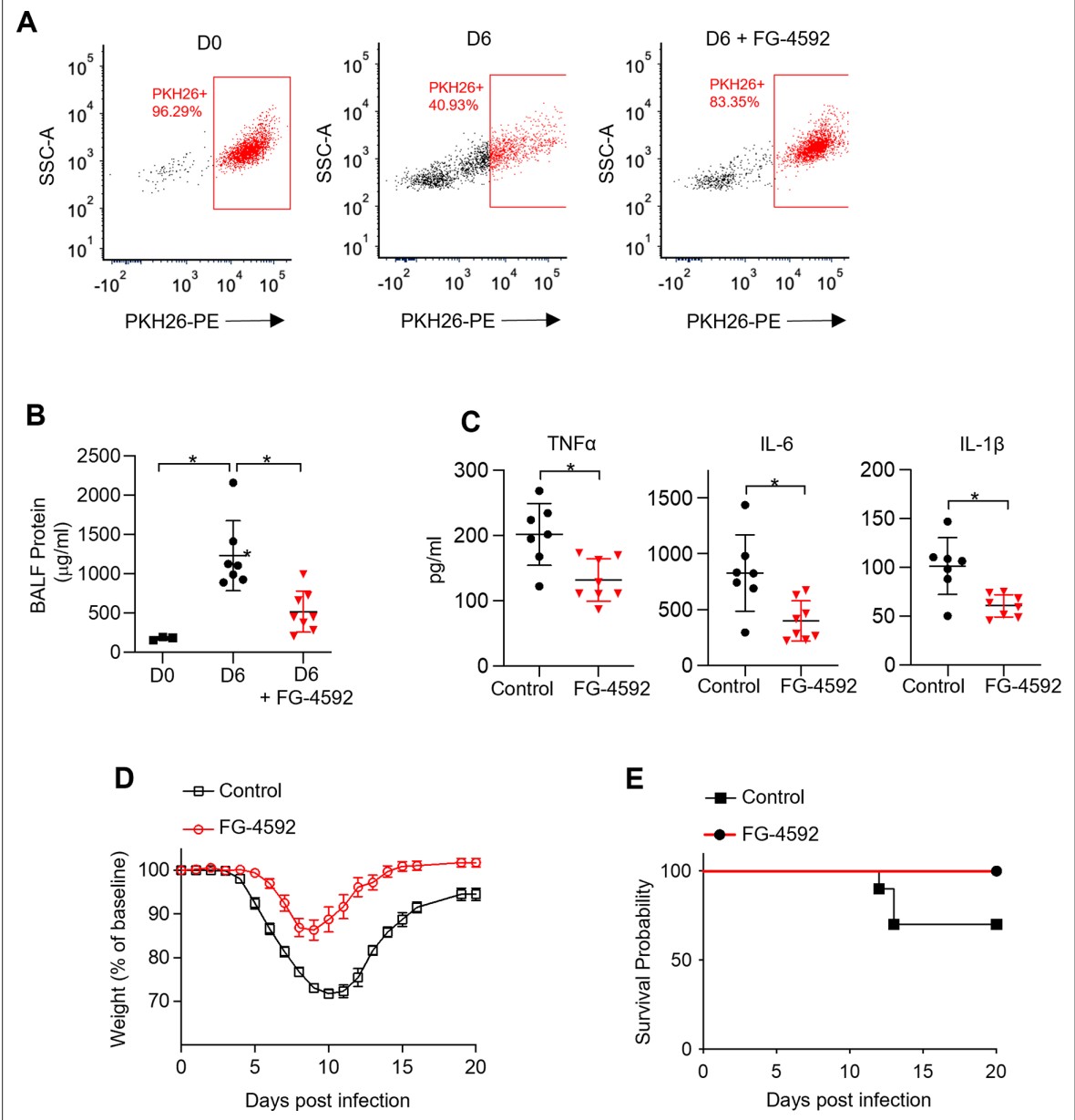

**Figure 7.** Non-hypoxic stabilization of hypoxia-inducible factor 1-alpha (HIF-1α) increases tissue-resident alveolar macrophage (TR-AM) survival and improves outcomes in influenza-induced acute lung injury. We intratracheally infected C57BL/6 mice with PR8 (100 PFU) and collected bronchoalveolar lavage fluid (BALF) on day 0 (D0) (uninfected) and day 6 (D6) post infection. Mice also received either the HIF-1α stabilizer (FG-4592) or vehicle control on D0. (**A**) Representative FACS plot of BALF macrophages. (**B**) BALF protein concentration. (**C**) BALF proinflammatory cytokine levels at D6. BALF data generated from two separate experiments (n = 7 mice/control group and n = 8 mice/FG-4592 group). BALF data significance was determined by unpaired, two-tailed *t*-test. (**D ,E**) C57BL/6 mice infected with PR8 (200 PFU) (10 mice/group). (**D**) Weight loss represented as percentage and normalized to D0. (**E**) Survival curve. All error bars denote mean ± SD. *p<0.05.

stabilization under normoxic conditions and BMDM HIF-1α expression and glycolytic output remained unchanged in response to hypoxia. Transcriptomic analysis found 741 DEGs in TR-AMs in response to hypoxia compared to only 260 DEGs in BMDMs. This differential sensitivity to hypoxia observed in TR-AMs and BMDMs is likely tied to differences in developmental processes and respective, local tissue environments (*Hussell and Bell, 2014*; *Bain and MacDonald, 2022*). TR-AMs are derived from embryonic progenitors (fetal monocytes) and mature upon lung localization during development (*Guilliams et al., 2013*). Under steady-state conditions, the high-oxygen, low-glucose environment of the alveolar lumen favors a cell type that relies on oxygen-based metabolism, making glycolysis

an inefficient means to fuel cellular processes. The transcriptional signature of TR-AMs is enriched with genes associated with lipid metabolism (*Gibbings et al., 2015*; *Leach et al., 2020*). The alveolar lumen contains an abundance of fatty acid molecules in the form of pulmonary surfactant. TR-AMs are critical in recycling surfactant in the healthy airway (*Whitsett et al., 2010*). While the TR-AMs' preferred steady-state mitochondrial fuel remains elusive, it is likely that they utilize fatty acids for energy by catabolizing pulmonary surfactant. ARDS leads to hypoxia accompanied by an influx of glucose and dramatic decreases in pulmonary surfactant molecules (*Hofer et al., 2015*; *Woods et al., 2016*; *Schousboe et al., 2022*). This suggests that TR-AMs would need to significantly augment their metabolism to survive and likely explains why HIF-1α induction in TR-AMs is so sensitive to hypoxia and why RNAseq analysis revealed a much greater number of hypoxia response genes in TR-AMs compared to BMDMs. In contrast, BMDMs/monocytes are derived from adult hematopoietic stems cells and arise from the bone marrow where oxygen concentrations vary from 1.5 to 4% (*Spencer et al., 2014*). BMDMs/monocytes circulate in the blood and traffic to sites of inflammation, which both have significantly higher glucose levels compared to the steady-state alveoli. These conditions favor constitutive expression of HIF-1α and necessitate a functional glycolytic phenotype to fuel cellular processes. Thus, macrophage/monocyte populations arising from the bone marrow, where hypoxic conditions are observed under steady-state conditions, are likely less sensitive to hypoxia stimulus in vitro.

Indeed, our in vivo experiments show that although TR-AMs are lost during ARDS, the surviving TR-AMs are glycolytically adapted and resemble recruited macrophages in metabolic gene expression. This is consistent with our previous findings that TR-AM viability is exquisitely sensitive to mitochondrial inhibition while BMDM viability is unaffected. Our current findings show that hypoxia restores viability to TR-AMs under mitochondrial inhibition and reverses the suppressive effects of this inhibition on TR-AM cytokine production supporting optimal effector function in a low-oxygen environment. These data suggest that the link between immune effector function and metabolism in TR-AMs is based on the overall cellular fitness, and that the environmental shift from high oxygen and low glucose under steady-state conditions to low oxygen and increased glucose during lung injury may necessitate cellular adaptation to the microenvironment to ensure survival.

While we are unable to directly measure the alveolar oxygen concentration during lung injury, several lines of evidence provide justification for the use of oxygen levels as low as 1.5% $O_2$ for our in vitro studies. Severe hypoxia is observed in the airways in both chronic and acute lung disease (*Schaible et al., 2010*). For example, bronchoscopic measurements in CF patients recorded $O_2$ levels below 1% in mucopurulent bronchi (*Worlitzsch et al., 2002*). Murine influenza infection yields a rapid decline in alveolar gas exchange in a manner that models human ARDS (*Wolk et al., 2008*; *Traylor et al., 2013*). These observations correlate with impaired alveolar fluid clearance, interstitial edema, alveolar damage, and inflammatory infiltrates that all likely limit oxygen diffusion into the alveolar space (*Aeffner et al., 2015*). In ARDS, cells such as TR-AMs residing in the areas of lungs that are filled with protein-rich fluid and those with complete atelectasis are potentially subjected to severe hypoxia and even anoxia. *Xi et al., 2017* used pimonidazole to identify hypoxic alveoli during influenza infection. While pimonidazole cannot provide quantitative measurements in relation to oxygen concentrations, biochemical analysis suggests that 2-nitroimidazole compounds label cells once $O_2$ levels drop to approximately 1.3% $O_2$ (*Gross et al., 1995*). Taken together, these observations suggest that the alveolar space can become severely hypoxic during influenza-induced lung injury.

Pharmacological depletion of TR-AMs has offered insight into their beneficial immunoregulatory properties in various lung injury models. TR-AMs have been shown to alleviate lung injury by clearing apoptotic neutrophils, suppressing T-cell-mediated inflammatory responses, and limiting dendritic cell infiltration and antigen presentation (*Thepen et al., 1989*; *Holt et al., 1993*; *Knapp et al., 2003*; *Jakubzick et al., 2006*). Loss of TR-AMs during influenza infection is known to enhance mortality, and several studies have shown that there is a progressive loss of TR-AMs over the time course of infection (*Kim et al., 2008*; *Ghoneim et al., 2013*; *Schneider et al., 2014*; *Cardani et al., 2017*; *Zhu et al., 2021*). Why TR-AMs are lost during the course of IAV infection is not understood. We hypothesized that failure to adapt to the hypoxic environment may play a role in the TR-AM loss. We found that HIF-1α activation was sufficient to promote glycolysis, and rescue TR-AM viability and effector function under mitochondrial inhibition. Furthermore, when mice were treated with a HIF-1α stabilizer at the time of influenza infection, TR-AM survival was increased and accompanied by reduced lung injury

and death. These findings suggest that HIF-1α is essential for TR-AM cell survival, and that increasing TR-AM cell number by promoting their adaptation to ARDS-associated changes in the microenvironment during infection reduces lung injury.

In agreement with our findings, Zhu and colleagues recently showed that TR-AMs at D6 of influenza infection have high HIF-1α expression; however, they suggested that HIF-1α stabilization in TR-AMs worsens lung injury through enhanced proinflammatory effector function (*Zhu et al., 2021*). This study relied on the use of a non-inducible *Cd11c* cre allele to knock out HIF-1α in TR-AMs. While TR-AMs do express high levels of Cd11c, this marker is not specific for TR-AMs and is also expressed in different monocyte/macrophage populations, dendritic cells, and natural killer cells (*Abram et al., 2014*). Moreover, as monocyte-derived macrophages enter the alveolar space, they begin to express Cd11c (*Misharin et al., 2013*). Thus, it cannot be determined from these experiments whether the observed effect of HIF-1α deletion on inflammation and lung injury was specific for TR-AMs as the recruited monocyte-derived macrophages will also lose HIF-1α as they enter the lungs. It is likely that the recruitment of monocyte-derived macrophages was inhibited by HIF-1α deletion, resulting in reduced inflammation and lung injury (*Cramer et al., 2003*). Using FG-4592 in vitro, we found that HIF-1α stabilization and glycolytic reprogramming resulted in no significant changes in proinflammatory cytokine production downstream of LPS. Furthermore, we show that treatment of mice with FG-4592 promoted survival of TR-AMs after influenza infection and led to reduced levels of lung injury. While intratracheal FG-4592 installation improved outcomes in influenza-infected mice, we cannot definitively say that the beneficial effects are solely related to HIF-1α induction in TR-AMs since our delivery strategy could also affect lung epithelial cells, which might indirectly contribute to the results. Methodologies allowing for TR-AM specific delivery of FG-4592 would resolve this issue. Alternatively, a genetic approach allowing for inducible HIF-1α activation could conclusively demonstrate whether HIF-1α expression specifically in TR-AMs promotes survival during lung injury since constitutive expression of HIF-1α leads to defects in TR-AM development (*Izquierdo et al., 2018*). There have also been no studies on the effects of long-term HIF-1α activation and resultant metabolic reprogramming in TR-AMs under steady state or during recovery from inflammatory stimuli. Thus, our short-term pharmacological activation of HIF-1α and glycolytic reprogramming in TR-AMs is a useful model for studying the role of metabolic reprogramming during lung injury.

Interestingly, we found that the effects of hypoxia on both TR-AM and BMDM effector function appear to be independent of HIF-1α. While HIF-1α stabilization was sufficient to drive metabolic reprogramming in TR-AMs, FG-4592 did not affect cytokine production downstream of LPS. By contrast, when cultured in hypoxia TR-AMs exhibited marked changes in their secreted cytokine profile including increased secretion of TNF-α, KC, and IL-1β and less IL-6 and CCL2 compared to normoxic controls. BMDMs exposed to hypoxia and treated with LPS responded similarly to TR-AMs with the exception to CCL2, which remained unchanged in response to hypoxia in BMDMs. This was independent of any effect of hypoxia on HIF-1α stabilization or target gene expression in BMDMs. Similarities between TR-AMs and BMDMs in terms of cytokine profiles would suggest that low oxygen concentrations may be driving these changes independently of HIF-1α stabilization. Further investigation will be required to elucidate the mechanisms by which a low-oxygen environment alters cytokine production.

In conclusion, under normoxic conditions, TR-AMs depend on mitochondrial respiration, inhibition of which leads to decreased effector response and cell death. Under hypoxic conditions, as occurs during ARDS, TR-AMs stabilize HIF-1α in a dose-dependent manner and have a robust HIF-1α response compared to BMDMs. Stabilization of HIF-1α allows TR-AMs to augment glycolytic function and prevents their death following the inhibition of mitochondrial respiration as well as following influenza A viral infection. These data suggest that therapies inducing HIF-1α in TR-AMs may be beneficial in ARDS by preventing their death through metabolic adaptation to the ARDS microenvironment that is low in $O_2$ and high in glucose.

## Materials and methods

**Key resources table**

| Reagent type (species) or resource | Designation | Source or reference | Identifiers | Additional information |
|---|---|---|---|---|
| Strain, strain background (*Mus musculus*) | C57BL/6J | Jackson Laboratory | Stock no. 000664 | 6–8 weeks |
| Strain, strain background (influenza A virus) | A/PR8/34 (H1N1) | BEI Resources, NIAID, NIH | NR-348 | |
| Antibody | Anti-HK2 (rabbit monoclonal) | Cell Signaling Technology | Cat# C64G5 | WB (1:1000) |
| Antibody | Anti-LDHA (rabbit polyclonal) | Cell Signaling Technology | Cat# 2012S | WB (1:1000) |
| Antibody | Anti-PHD2/EGLN1 (rabbit monoclonal) | Cell Signaling Technology | Cat# 4835 | WB (1:1000) |
| Antibody | Anti- PHD3/EGLN3 (rabbit polyclonal) | Novus Biologicals | Cat# NB100-303 | WB (1:1000) |
| Antibody | Anti-IL-1β (mouse monoclonal) | Cell Signaling Technology | Cat# 12242 | WB (1:1000) |
| Antibody | Anti-Lamin B1 (rabbit polyclonal) | ProteinTech | Cat# 12987-1-AP | WB (1:1000) |
| Antibody | Anti-HIF-1α (rabbit polyclonal) | Cayman Chemical | Cat# 10006421 | WB (1:500) |
| Antibody | Anti-α-Tubulin (mouse monoclonal) | Sigma | Cat# T6074 | WB (1:20,000) |
| Antibody | Anti-rabbit IgG, HRP-linked Antibody (goat polyclonal) | Cell Signaling Technology | Cat# 7074 | WB (1:2500) |
| Antibody | Anti-mouse IgG, HRP-linked Antibody (horse polyclonal) | Cell Signaling Technology | Cat# 7076 | WB (1:2500) |
| Antibody | CD16/CD32 (FcBlock) (rat monoclonal) | BD Biosciences | Clone 2.4G2; Cat# 553141 | Flow cytometry (1:50) |
| Antibody | Alexa Fluor 700 anti-mouse Ly-6G (rat monoclonal) | BioLegend | Clone 1A8; Cat# 553141 | Flow cytometry (1:250) |
| Chemical compound, drug | FG-4592 (roxadustat) | Cayman Chemical | Cat# 15294 | |
| Chemical compound, drug | Recombinant mouse M-CSF | BioLegend | 576406 | |
| Chemical compound, drug | Oligomycin | Fisher Scientific | 49-545-510MG | |
| Chemical compound, drug | FCCP | MilliporeSigma | C2920 | |
| Chemical compound, drug | Antimycin A | MilliporeSigma | A8674 | |
| Chemical compound, drug | Rotenone | MilliporeSigma | R8875 | |
| Chemical compound, drug | Lipopolysaccharide | Santa Cruz | sc-3535 | |
| Commercial assay or kit | Mouse IL-6 DuoSet ELISA | R&D Systems | DY406 | |
| Commercial assay or kit | Mouse TNF-α DuoSet ELISA | R&D Systems | DY410 | |
| Commercial assay or kit | Mouse KC DuoSet ELISA | R&D Systems | DY453 | |
| Commercial assay or kit | Mouse CCL2 DuoSet ELISA | R&D Systems | DY479 | |
| Commercial assay or kit | Mouse IL-1β alpha DuoSet ELISA | R&D Systems | DY401 | |
| Commercial assay or kit | Lactate Assay Kit | MilliporeSigma | MAK064-1KT | |
| Commercial assay or kit | Mouse Macrophage Nucleofector Kit | Lonza | VPA-1009 | |
| Commercial assay or kit | Seahorse XFe24 FluxPak | Agilent | 102340-100 | |
| Commercial assay or kit | NE-PER Nuclear and Cytoplasmic Extraction Reagents | Thermo Fisher | Cat# 78833 | |
| Other | PKH26 Cell Linker Dye for Phagocytic Cell Labeling | MilliporeSigma | Cat# PKH26PCL-1KT | Dye to distinguish between TR-AMs and Mo-AMs |
| Other | SYTOX Green Nucleic Acid Stain | Thermo Fisher | Cat# S7020 | Stain to distinguish between live and dead cells. |
| Sequence-based reagent | *Rpl19*_F | This paper | PCR primers | CCGACGAAAGGGTATGCTCA |
| Sequence-based reagent | *Rpl19*_R | This paper | PCR primers | GACCTTCTTTTTCCCGCAGC |
| Sequence-based reagent | *Il6*_F | This paper | PCR primers | TTCCATCCAGTT GCCTTCTTGG |
| Sequence-based reagent | *Il6*_R | This paper | PCR primers | TTCCTATTTCCA CGATTTCCCAG |

| Reagent type (species) or resource | Designation | Source or reference | Identifiers | Additional information |
|---|---|---|---|---|
| Sequence-based reagent | Tnfa_F | This paper | PCR primers | AGGGGATTAT GGCTCAGGGT |
| Sequence-based reagent | Tnfa_R | This paper | PCR primers | CCACAGTCCAGGTCACTGTC |
| Sequence-based reagent | Il1b_F | This paper | PCR primers | GCCACCTTTT GACAGTGATGAG |
| Sequence-based reagent | Il1b_R | This paper | PCR primers | GACAGCCCA GGTCAAAGGTT |
| Sequence-based reagent | Kc_F | This paper | PCR primers | AGACCATGGC TGGGATTCAC |
| Sequence-based reagent | Kc_R | This paper | PCR primers | ATGGTGGCTATGACTTCGGT |
| Sequence-based reagent | Ccl2_F | This paper | PCR primers | CTGTAGTTTTT GTCACCAAGCTCA |
| Sequence-based reagent | Ccl2_R | This paper | PCR primers | GTGCTGAAGA CCTTAGCCCA |
| Sequence-based reagent | Non-targeting (control) siRNA | Dharmacon | D-001810-01 | |
| Sequence-based reagent | Hif1a #1; J-040638-06 | Dharmacon | J-040638-06 | |
| Sequence-based reagent | Hif1a #2; J-040638-07 | Dharmacon | J-040638-07 | |
| Software, algorithm | FastQC | Babraham Institute | RRID:SCR_014583 | |
| Software, algorithm | STAR | PMID:23104886 | RRID:SCR_015899 | |
| Software, algorithm | DESeq2 | Bioconductor | RRID:SCR_015687 | |
| Software, algorithm | Reactome Cytoscape Plugin | PMID:14597658 | RRID:SCR_003032 | |
| Software, algorithm | Prism 9 | GraphPad | RRID:SCR_002798 | |

## Primary culture of macrophages

All animal experiments and procedures were performed according to the protocols (ACUP7236 and ACUP72484) approved by the Institutional Animal Care and Use Committee at the University of Chicago. 6–8-week-old C57BL/6 mice were humanely euthanized, and their TR-AMs were isolated via standard bronchoalveolar lavage (intratracheal instillation) using PBS + 0.5 mM EDTA. Following isolation, TR-AMs were counted, plated in RPMI 1640 (Thermo Fisher, Cat# 11875119) supplemented with 10% FBS (Gemini, Cat# 100-106) and 1% penicillin-streptomycin (Gemini, Cat# 400-109), and allowed to adhere to tissue culture plates for 1 hr prior to experimentation. BMDMs were generated by isolating bone marrow cells from the femur and tibia bones of 6–8-week-old C57BL/6 mice. Bone marrow cells were differentiated into BMDMs using 40 ng/ml recombinant M-CSF (BioLegend, Cat# 576406) in the same media formulation as TR-AMs. On day 7, BMDMs were replated and allowed to adhere to tissue culture plates for 2 hr prior to experimentation. After adherence, cells were washed, fresh media added, and placed under experimental conditions. For inflammatory stimulation, LPS was used at a concentration of 20 ng/ml. For hypoxia experiments, macrophage cultures were placed in an airtight incubator system that utilizes $N_2$ displacement of $O_2$ to achieve hypoxic conditions (Coy Hypoxic Chamber- $O_2$ Control InVitro Glove Box). The sealed system ensures minimal fluctuations in $O_2$ levels in experiments when treating cultures and collecting samples under hypoxic conditions.

## Bioenergetic measurements

Glycolytic and mitochondrial respiration rates were measured using the XFe24 Extracellular Flux Analyzer (Agilent, Santa Clara, MA). BMDMs and TR-AMs were seeded at $4.0 \times 10^4$/well onto Seahorse XF24 Cell Culture Microplates. Cells were equilibrated with XF Base media (Agilent, Cat# 103334-100) at 37°C for 30 min in the absence of $CO_2$. Glycolytic rate was assessed using the manufacturer's protocol for the Seahorse XF Glycolysis Stress Test followed by sequential injections with glucose (10 mM), oligomycin (1.0 μM), and 2-DG (100 mM). Mitochondrial respiration rate was measured using the Seahorse XF Mito Stress Test according to the manufacturer's protocol followed by sequential injections with oligomycin (1.0 μM), FCCP (1.0 μM for BMDMs and 4.0 μM for TR-AMs), and rotenone/antimycin A (1.0 μM). Assessment of real-time metabolic responses to LPS was performed

using the protocol detailed in an application note provided by Agilent (*Kam Y and Dranka, 2017*). In brief, following plating, cells were equilibrated in XF base media supplemented with 10 mM glucose, 2 mM L-glutamine, 1 mM sodium pyruvate (Sigma, Cat# 11360070) and 5 mM HEPES (Sigma, Cat# 15630080), pH 7.4, and incubated at 37°C without $CO_2$ for 30 min prior to XF assay. Baseline metabolic rates were measured followed by direct injection of LPS (final concentration: 20 ng/ml). Bioenergetic rates were subsequently measured every 3 min for approximately 5 hr in total.

Due to the limitations of the XFe24 Extracellular Flux Analyzer, all bioenergetic analyses on hypoxic samples were performed in the following manner. Cells were treated under hypoxic conditions (most commonly for 16 hr), then bioenergetic analysis was performed under normoxic conditions. Moreover, it is impossible to evaluate varying levels of hypoxia on a single Seahorse microplate. Thus, energy curves comparing varying levels of $O_2$ (i.e., *Figure 1A*) were performed individually and then subsequently represented on the same graph for comparison. All individual experiments were repeated a minimum of three times to ensure accurate representation and statistical comparison.

## Cell lysis, subcellular fractionalization, and immunoblotting

Whole-cell lysates were prepared by scraping cells into lysis buffer containing 25 mM Tris HCl (pH 7.6), 150 mM NaCl, 1% NP-40, 1% sodium deoxycholate, 0.1% SDS, 0.1% Benzonase, and Halt Protease Inhibitor Cocktail (Thermo Fisher, Cat# 1861284 and 78430). Samples were centrifuged at 16,000 × *g* at 4°C for 5 min to pellet cellular debris. Subcellular fractionalization and lysate preparation were carried out using the NE-PER Nuclear and Cytoplasmic Extraction Reagents (Thermo Fisher, Cat# 78833). Lysate protein concentration was determined using the Pierce BCA Protein Assay Kit (Thermo Fisher, Cat# 23225). Equal concentrations of samples (15 μg for whole-cell lysates and 5 μg for nuclear fractions) were resolved on Criterion gels (Bio-Rad, Cat# 5671093 and 5671094) and transferred to nitrocellulose (Bio-Rad, Cat# 1620167). Primary antibodies used were rabbit anti-HK2 (Cell Signaling, Cat# C64G5, 1:1000), rabbit anti-LDHA (Cell Signaling, Cat# 2012, 1:1000), rabbit anti-PHD2/Egln1 (Cell Signaling, Cat# 4835, 1:1000), rabbit anti-Egln3/PHD3 (Novus Biologicals, Cat# NB100-303, 1:1000), mouse anti-IL1β (Cell Signaling, Cat# 12242 1:1000), rabbit anti-Lamin B1 (ProteinTech, Cat# 12987-1-AP, 1:1000), rabbit anti-HIF-1α (Cayman Chemical, Cat# 10006421, 1:500), and mouse anti-tubulin (Sigma, Cat# T6074, 1:20,000). Secondary antibodies used were anti-rabbit IgG HRP-linked antibody (Cell Signaling, Cat# 7074, 1:2500) and goat anti-mouse IgG HRP-linked antibody (Cell Signaling, Cat# 7076, 1:2500). Protein expression was visualized using Immobilon ECL Ultra Western HRP Substrate (MilliporeSigma, Cat# WBULS0500) in combination with the Bio-Rad ChemiDoc Touch Imaging system. All immunoblot data were repeated in at least three independent experiments.

## Quantitative PCR

RNA was isolated from cells using the Direct-zol RNA MiniPrep kit (Zymo Research, Cat# R2052) and reverse transcribed using iScript Reverse Transcription Supermix (Bio-Rad, Cat# 1708841). Quantitative mRNA expression was determined by real-time qRT-PCR using iTaq Universal SYBR Green Supermix (Bio-Rad, Cat# 172-5121). *rlp19* served as a housekeeping gene, and gene expression was quantified using the ΔΔct method to determine relative fold change (FC). The following mouse-specific primer sequences were used: *Rlp19* (5'-CCGACGAAAGGGTATGCTCA-3', 5'-GACCTTCTTTTT CCCGCAGC-3'), *Il6* (5'-TTCCATCCAGTTGCCTTCTTGG-3', 5'-TTCCTATTTCCACGATTTCCCAG-3'), *Tnfa* (5'-AGGGGATTATGGCTCAGGGT-3', 5'-CCACAGTCCAGGTCACTGTC-3'), *Il1b* (5'-GCCACCTT TTGACAGTGATGAG, 5'-GACAGCCCAGGTCAAAGGTT-3'), *Kc* (5'-AGACCATGGCTGGGATTCAC -3', 5'-ATGGTGGCTATGACTTCGGT-3'), and *Ccl2* (5'-CTGTAGTTTTTGTCACCAAGCTCA-3', 5'-GTGC TGAAGACCTTAGCCCA-3').

## RNA-sequencing

RNA was isolated and submitted to the University of Chicago Genomics Core Facility for sequencing with the Illumina NovaSEQ6000 sequencer (100 bp paired-end). Sequencing read (FASTQ) files were generated and assessed for per base sequence quality using FastQC. Reads were mapped to the mouse genome (GRCm38.p6, GENCODE) using Spliced Transcripts Alignment to a Reference (STAR) software, and the resulting gene transcripts were quantified using featureCounts.

Gene counts were then imported into R for differential expression analysis using the Bioconductor package DESeq2. Gene counts were filtered to remove low-expressing genes at a threshold of 2

counts per million. Differential expression was calculated between normoxia and hypoxia groups for both AMs and BMDMs. Differential gene expression was considered significant for genes with a false discovery rate (FDR)-adjusted p-value <0.05 and FC > 2. Reactome enrichment hit pathways and the linked gene lists from significant DEGs were identified by using Reactome Cytoscape Plugin (*Shannon et al., 2003*; *Wu et al., 2010*). Oxidative phosphorylation and glycolysis gene sets were extracted from UniProtKB, then their associated DEG read counts (TPM) were normalized using gensvm R package gensvm.maxabs.scale function and center scaled for heatmap visualization. All heatmaps were generated with Pretty heatmaps R package pheatmap function.

## Cytokine analysis

Secreted TNFα, IL-6, KC, CCl2, and IL-1β levels were evaluated in macrophage media using a standard sandwich ELISA (R&D Systems DuoSet ELISA Development System, Cat# DY410, DY406, DY453, DY479, and DY401). For IL-1β sample collection, 5 mM ATP was added to TR-AM cultures for 30 min following 6 hr LPS treatment to activate caspase 1, ensuring proIL-1β cleavage and IL-1β release. Rotenone and antimycin A concentrations were 20 nM for TR-AMs and 1 μM for BMDMs when used in ELISA experiments.

## Metabolomics

TR-AMs were plated at $2.5 \times 10^6$ on 60 mm plates for metabolite extraction. Following treatment, cells were washed twice with a 5% mannitol solution and metabolites were extracted using 400 μl 100% methanol. 275 μl of aqueous internal standard solution was mixed in with the methanol and the extract solution was transferred to a microcentrifuge. The extracts underwent centrifugation at 2300 $\times$ g at 4°C for 5 min to precipitate insoluble material and the resulting supernatant was transferred to centrifugal filter units (Human Metabolome Technologies [HMT], Boston, MA). Filtering of supernatant occurred at 9100 $\times$ g at 4°C for 2 hr. The filtrate was sent to HMT and analyzed using capillary electrophoresis–mass spectrometry.

## Sulforhodamine B (SRB) colorimetric assay

In vitro cytotoxicity was measured using the SRB assay (*Vichai and Kirtikara, 2006*). Following treatment, cells were fixed in 10% TCA and then stained with SRB dye. Cellular protein-dye complexes were solubilized in 10 mM Tris base and the samples were read at OD 510 using a microplate reader. Data was normalized to the untreated, normoxia groups, which were representative of no cellular damage. ETC inhibitor concentrations in BMDMs were as follows: 1 μM rotenone and 1 μM antimycin A. ETC inhibitor concentrations in TR-AMs were as follows: 500 nM rotenone and 100 nM antimycin A.

## SiRNA knockdown

SiRNA knockdown was performed using the Amaxa Mouse Macrophage Nucleofector Kit (Lonzo, Cat# VPA-1009). $1.0 \times 10^6$ cells/reaction were resuspended in transfection solution with siRNA of interest (Dharmacon, Non-Targeting Control siRNA: D-001810-01; mouse *Hif1a* siRNA #1; J-040638-06; mouse *Hif1a* siRNA #2; J-040638-07). The cell solution was then subjected to electroporation (Lonza Nucleofector 2b Electroporator: Setting Y-001). Cells were plated and allowed to rest for 6 hr, then subjected to normoxia or hypoxia for 16 hr.

## Lactate assay

Secreted lactate was measured using the lactate colorimetric assay kit (Sigma, Cat# MAK064-1KT). Cells cultured in serum-free DMEM media (RPMI media and serum interfere with assay) and exposed to normoxia or 1.5% $O_2$. Samples were collected at 16 hr post treatment and manufacturer's protocol was followed to measure lactate.

## Murine influenza infection protocol and survival studies

C57BL/6 mice (6–8 weeks old) were anesthetized and challenged intratracheally (IT) with mouse-adapted influenza (A/PR8/34; 200 plaque-forming units [PFU]). A single FG-4592 (50 μM) treatment was administered (IT) simultaneously with IAV. Body weight and survival was monitored every 24 hr for 20 days (10 mice/group). Body weight is represented as percent deviation from baseline at time of

infection. 'Influenza A virus, A/PR8/34 (H1N1), NR-348' was obtained through BEI Resources, NIAID, NIH.

## BALF analysis

C57BL/6 mice were euthanized and a single 0.5 ml saline wash was instilled into the lungs via the trachea and subsequently collected. BALF protein concentration was determined using the Pierce BCA Protein Assay Kit (Thermo Fisher, Cat# 23225). BALF TNFα, IL-6, and IL-1β were measured using sandwich ELISA.

## Flow cytometry

C57BL/6 mice (6–8 weeks old) were anesthetized with isoflurane and underwent retro-orbital injection with 100 µl PKH26 Red Fluorescent Cell Linker Dye for Phagocytic Cell Labeling (Cat# PKH26PCL-1KT; MilliporeSigma) 1 day before lung challenge. The mice were then challenged intratracheally with IAV 100 (PFU). FG-4592 (50 µM) was administered intratracheally at the same time of PR8 infection. After challenge, the mice were euthanized and immune cells were collected via BAL. BAL cells were first treated with Fc Block (clone 2.4G2, Cat# 553141; BD Biosciences) and stained with fluorochrome-conjugated antibodies. The antibodies used were Alexa Fluor 700 anti-mouse Ly-6G (clone 1A8, Cat# 127621, 1:250; BioLegend). Immediately before sorting, cells were resuspended in sorting buffer (0.2% BSA in PBS) containing 5 nM SYTOX Green Nucleic Acid Stain (Cat# S7020; Thermo Fisher) to distinguish between live and dead cells. Cell sorting was performed on a FACS Aria II instrument, and data were acquired using BDFACS Diva software and analyzed with FCS Express 7 software. First, debris, red blood cells, and lymphocytes were eliminated based on size (FSC) and granularity (SSC). Next gates selected for live cells (FITC−) and eliminated neutrophils (Ly6G+). Based on previous validation experiments, the remaining cells are of macrophage lineage with TR-AMs being PKH26+ (SiglecF+,F4/80+,Cd11c$^{Hi}$, Ly6c$^{Lo}$) and Mo-AMs being PKH26− (F4/80+,Cd11c$^{Lo}$, Ly6c$^{Hi}$). PKH26+ and PKH26− cells were sorted into RNA lysis buffer and samples were prepared for RNAseq.

## Statistics

The data were analyzed in Prism 9 (GraphPad Software Inc). All data are shown as mean ± standard deviation (SD). Significance was determined by unpaired, two-tailed Student's $t$-test for comparisons between two samples or by ANOVA using Bonferroni correction for multiple comparisons. p-Values <0.05 were considered statistically significant.

## Acknowledgements

This work was supported by grants T32HL007605 (PSW, LMK, ORS, GMM), R01HL151680 (RBH) and R01ES010524, U01ES026718, P01HL144454, and Department of Defense W81XWH-16-1-0711 (GMM).

# Additional information

## Competing interests

Parker S Woods, Robert B Hamanaka, Gökhan M Mutlu: has a pending patent on targeting tissue-resident alveolar macrophage metabolism to prevent their death during ARDS. (ARCD.P0740US. P1/1001176943). The other authors declare that no competing interests exist.

## Funding

| Funder | Grant reference number | Author |
| --- | --- | --- |
| U.S. Department of Defense | W81XWH-16-1-0711 | Gökhan M Mutlu |
| National Institute of Environmental Health Sciences | R01ES010524 | Gökhan M Mutlu |

| Funder | Grant reference number | Author |
|---|---|---|
| National Heart, Lung, and Blood Institute | R01HL151680 | Robert B Hamanaka |
| National Institute of Environmental Health Sciences | U01ES026718 | Gökhan M Mutlu |
| National Heart, Lung, and Blood Institute | P01HL144454 | Gökhan M Mutlu |
| National Heart, Lung, and Blood Institute | T32HL007605 | Parker S Woods<br>Lucas M Kimmig<br>Obada R Shamaa<br>Gökhan M Mutlu |

The funders had no role in study design, data collection and interpretation, or the decision to submit the work for publication.

## Author contributions

Parker S Woods, Conceptualization, Data curation, Formal analysis, Validation, Writing – original draft, Writing – review and editing; Lucas M Kimmig, Data curation, Formal analysis, Validation, Writing – review and editing; Kaitlyn A Sun, Data curation, Formal analysis, Writing – review and editing; Angelo Y Meliton, Data curation, Formal analysis, Validation, Investigation, Writing – review and editing; Obada R Shamaa, Data curation, Investigation, Writing – review and editing; Yufeng Tian, Data curation, Validation, Investigation; Rengül Cetin-Atalay, Formal analysis, Investigation, Writing – original draft, Writing – review and editing; Willard W Sharp, Resources, Methodology, Writing – review and editing; Robert B Hamanaka, Conceptualization, Formal analysis, Supervision, Investigation, Writing – original draft, Writing – review and editing; Gökhan M Mutlu, Conceptualization, Resources, Supervision, Funding acquisition, Investigation, Writing – original draft, Project administration, Writing – review and editing

## Author ORCIDs

Parker S Woods http://orcid.org/0000-0002-9054-4196
Robert B Hamanaka http://orcid.org/0000-0002-8909-356X
Gökhan M Mutlu http://orcid.org/0000-0002-2056-612X

## Ethics

All animal experiments and procedures were performed according to the protocols (ACUP7236 and ACUP72484) approved by the Institutional Animal Care and Use Committee at the University of Chicago.

## Decision letter and Author response

Decision letter https://doi.org/10.7554/eLife.77457.sa1
Author response https://doi.org/10.7554/eLife.77457.sa2

# Additional files

## Supplementary files

• Transparent reporting form

## Data availability

Source Data files have been provided for Figures 2C, and 5B, C.

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
