## [Editor Report]

This work adds to an already abundant literature demonstrating that TR-AMs are a phenotypically and functionally distinct population of macrophages. Work in this article is the first to characterize the effect of hypoxia on metabolic and inflammatory responses in TR-AMs vs macrophages derived from other sites (bone marrow).

---

## [Decision Letter]

**Decision letter after peer review:**

Thank you for submitting your article "HIF-1α induces glycolytic reprogramming in tissue-resident alveolar macrophages to promote cell survival during acute lung injury" for consideration by *eLife*. Your article has been reviewed by 2 peer reviewers, and the evaluation has been overseen by a Reviewing Editor and Paul Noble as the Senior Editor. The reviewers have opted to remain anonymous.

Essential revisions:

Both reviewers comment on the merit of the study but would like some additional experiments related to the targeting of HIF-1.

1) Perform in vitro studies to validate findings with pharmacological inhibitors of HIF1.

2) Provide justification for the oxygen concentrations selected in the study.

3) Combine the first 2 figures.

*Reviewer #1 (Recommendations for the authors):*

Suggestions:

1) Use of complementary genetic approaches to manipulating HIF1 would strengthen the manuscript.

2) Justification for the use of 1.5 and 3% oxygen is important. It is not clear this concentration of oxygen is relevant to the alveolar compartment.

3) Exploring mechanisms linking HIF1 to TR-AM survival would strengthen the investigation.

*Reviewer #2 (Recommendations for the authors):*

I believe these findings are appropriate for a publication in eLife. I have a few comments.

- Can the authors argue what the difference is between BMDM and TRAM that causes the differential response to hypoxia. It may be due to tissue of origin but this requires an extensive discussion.

- The most interesting part of the manuscript in my opinion is the mouse study. It is impressive that FG drugs can extend survival. The causal link between macrophages and drug-induced survival is missing. It is unclear whether mice survive due to the effect of the FG drug on macrophages or some other cell type. The precise way of doing this experiment is to activate HIF specifically in TRAM and repeat the infection experiment. While I think this experiment is important, I think it is unreasonable to ask for this experiment so the authors should at least mention this potential limitation.

- The paper can benefit from a more compact presentation. The first 2 figures can be combined. Additionally, in my opinion, the RNAseq data on TRAMs isolated from infected mice should come after the functional data.

---

## [Author Response]

1) Perform in vitro studies to validate findings with pharmacological inhibitors of HIF1.

As recommended, we performed additional in vitro studies using siRNA technology to knockdown HIF-1a in macrophages. These results are included in Figure 1—figure supplement 1.

2) Provide justification for the oxygen concentrations selected in the study.

We expanded the Discussion section to provide justification for the oxygen concentration we used in our studies.

3) Combine the first 2 figures.

As recommended, we combined figures 1 and 2.

Reviewer #1 (Recommendations for the authors):Suggestions:1) Use of complementary genetic approaches to manipulating HIF1 would strengthen the manuscript.

We agree with the concerns about the specificity of echinomycin and other pharmacologic inhibitors of HIF-1a. As recommended, we used a genetic approach with siRNA to knockdown HIF1a expression and performed experiments to complement the initial data we generated using echinomycin. These new data confirmed our initial results with echinomycin and are included in the revised manuscript.

2) Justification for the use of 1.5 and 3% oxygen is important. It is not clear this concentration of oxygen is relevant to the alveolar compartment.

Cells such as TR-AMs residing in the areas of lungs that are filled with protein rich ARDS fluid and those atelectatic may potentially be subjected to severe hypoxia and even anoxia. While we are unable to directly measure the alveolar oxygen concentration during lung injury, murine influenza infection yields a rapid decline in alveolar gas exchange in a manner that models human ARDS causing severe hypoxia. Using pimonidazole investigators have shown that influenza infection in mice can lead to local O2 concentration that are less than 1.5%. Taken together, these observations suggest that the alveolar space can become severely hypoxic during influenza-induced lung injury. We revised the discussion to provide justification for the use of 1.5% O2.

3) Exploring mechanisms linking HIF1 to TR-AM survival would strengthen the investigation.

Our results suggest that metabolic flexibility provided by HIF-1alpha improve survival of TR-AMs, which are susceptible to the inhibition of mitochondrial respiration. However, we agree with the reviewer that there might be additional mechanisms that contribute to the survival benefit. We included this as a limitation of our study.

Reviewer #2 (Recommendations for the authors):I believe these findings are appropriate for a publication in eLife. I have a few comments.– Can the authors argue what the difference is between BMDM and TRAM that causes the differential response to hypoxia. It may be due to tissue of origin but this requires an extensive discussion.

We agree with the reviewer that the environment in which BMDMs and TR-AMs reside plays a role in their metabolic phenotype. We expanded the discussion section to emphasize this point.

– The most interesting part of the manuscript in my opinion is the mouse study. It is impressive that FG drugs can extend survival. The causal link between macrophages and drug-induced survival is missing. It is unclear whether mice survive due to the effect of the FG drug on macrophages or some other cell type. The precise way of doing this experiment is to activate HIF specifically in TRAM and repeat the infection experiment. While I think this experiment is important, I think it is unreasonable to ask for this experiment so the authors should at least mention this potential limitation.

We thank and agree with the reviewer’s comment that FG compound will likely affect other cells (alveolar epithelial cells) in the lungs and therefore we cannot exclude the possibility that HIF activation with FG in epithelial cells contributing to the beneficial effects of HIF activation. Unfortunately, Cd11c conditional knockout mice are not commercially available and mouse models that promote constitutive Cd11c HIF-1a expression lead to negative alterations in TR-AM maturation. As the reviewer stated we will need to develop a murine model in which HIF is conditionally activated only in TR-AMs, which is a limitation that we now acknowledge in the revised manuscript.

– The paper can benefit from a more compact presentation. The first 2 figures can be combined. Additionally, in my opinion, the RNAseq data on TRAMs isolated from infected mice should come after the functional data.

We combined figures 1 and 2 as recommended by the reviewer.